# Review Article: a systematic literature review of research trends and authorships on natural hazards, disasters, risk reduction and climate change in Indonesia

Riyanti Djalante [1,2,3]

[1]United Nations University – Institute of Environment and human Security, Bonn, 53117, Germany
[2] United Nations University – Institute for the Advanced Study of Sustainability, Tokyo, 150-8925, Japan (Current Affiliation)
[3] University of Halu Oleo, Kendari, 93111, Sulawesi Tenggara, Indonesia

*Correspondence to*: Riyanti Djalante (djalante@unu.edu)

**Abstract.**

Indonesia is one of the most vulnerable countries from disasters and climate change. While there has been a proliferation of academic publications on natural hazards, risks, and disasters on Indonesia, there has not yet a systematic literature review (SLR) to determine the progress, key topics, authorships and directions for further research. SLR is important so researchers can build upon existing works, avoid bias, determine major research, need for further research and to strengthen research
capacity in the future. The author conducts a SLR of publications indexed within the Scopus database from 1900 to 2016 on topics related to disasters and climate change in Indonesia. Two major findings are outlined. The first is related to major research topics: (1) natural hazard, risk and disaster assessments (HRD), (2) disaster risk reduction (DRR), and (3) climate change risks, vulnerability, impacts and adaptation (CC). More than half of the publications are related to HRD and focus on volcanic eruptions, tsunami and earthquakes. Publications on DRR are related to governance, early warning systems, and
recovery and reconstruction. Those on CC discuss carbon emission, forestry, governance, and sectoral impacts. The author calls for future research on different hazards, different locations, and impacts of disasters and climate change. Risks and vulnerability assessments from both hydro-meteorological and geophysical hazards are needed. Other locations beyond Sumatera and Java are to be examined. Urban risk assessments, and the economic and social impacts of disasters and climate change on the urban communities are equally important. Risk governance at the national, local and community level are to be
strengthened to increase resilience. The second finding presents the roles of Indonesian researchers and organizations. Findings show limited progress in research, publication and collaboration. International/ non-Indonesian authors dominate the literature and only half of the publications are co-authored by Indonesians. International collaborations have been conducted by very few Indonesian organizations. These could be due to limited experience in academic collaboration, power play amongst researchers, lack of research capacity, weak English academic writings skills, and limited incentives within higher education
system. The author recommends more funding and incentives for collaborations, training on English academic writing and journal article publications, capacity building for early career, female and social science researchers, encourages multi-disciplinary collaborations, and strengthening of science communication to social media outlets and science-policy advocacy.

# 1 Introduction

Disasters and their associated social and economic impacts are on the rise (EMDAT, 2018). The last decade has witnessed the highest number and impacts from disasters and 2015-2017 have been the hottest years ever (WMO, 2017). The Asia Pacific region has experienced the highest number of disasters (EMDAT, 2017), within which Indonesia is one of the most at risk countries to disasters and climate change (EMDAT, 2017). Between the period of 1900 to 2017, there have been a total of 489 disasters in Indonesia caused by natural hazards, with almost 242 thousand deaths, 30.7 million people affected and total damage almost 30 Billion USD (EMDAT, 2017). Geophysical disasters caused more than 95% deaths while the hydrological, meteorological and climatological disasters occur more frequently, affected more people, and caused more damages (EMDAT, 2017). To address this, the Sendai Framework for Disaster Risk Reduction (SFDRR) calls for multi hazard, integrated and inclusive approach for DRR and climate change adaptation (CCA) (UN/ISDR, 2015).

Studies on disasters have expanded enormously globally which calls for frequent synthesis of the research trends and topics, issues, challenges and strategies and innovations in dealing with those disasters. The role of science in influencing DRR policy is recognized and studies are needed to identify key lessons learnt and policy effectiveness. There is also call for giving more voices and strengthening capacities of local scientists in contributing to the generation of knowledge. It is often the case that local scientists are being left out in the internationally research collaborations and publications (Elsevier, 2017). The global progress on scholarly publications on disaster science is documented only recently by Elsevier in a report 'A Global Outlook on Disaster Science (Elsevier, 2017). It looks at scholarly outputs and impacts of disaster science according to the SFDRR, and documents progress of productivity of countries in in producing scholarly studies on disasters. There are 27, 273 scholarly outputs which represent only 0.22% of the world's total output, with countries such as China, United States of America and Japan dominate. Indonesia is part of countries that produces more specialized outputs in disaster science than global average (Elsevier, 2017). A detailed study that looks at progress of research and roles of researchers in Indonesia is not yet available. This paper aims to systematically review literature related to natural hazards and risks, DRR, and climate change vulnerability, impact, and assessments in Indonesia. A systematic literature review (SLR) is a method for systematically reviewing evidence or literature with explicit and transparent methods (Gill and Malamud, 2014). SLR enables determination of topics that have been heavily researched, build upon others´ existing works, avoid bias and repeat heavily researched topics (Khan et al., 1996). It is important to gauge when, how and by whom the research has been conducted so that future strategies for strengthening research capacity can be recommended (Mallett et al., 2012). There are two research objectives adopted. *First* is to determine progress of research in natural hazards, risks, disasters and climate change in Indonesia within the timeframe from 1900 to 2016. *Second* is to examine roles and progress of Indonesian authors in contributing to research, international publications and collaborations. The structure of this paper is as follows. The first section presents the rationale, aim and objectives. Next it outlines the research method. The third section presents results and discussions on key research topics and timelines, and progress of Indonesian researchers and organizations. The last section outlines recommendations to increase the quality of publications and scientific collaborations in international spheres, along with policy-relevant recommendations.

## 2 Research method

### 2.1 Data collection and multi-stage processes

The SLR method has been used in the fields of health (e.g. Moher et al., 2009a), software engineering (e.g. Kitchenham et al., 2009), and engineering (e.g. Gosling and Naim, 2009). There have also been studies that use this form of review in topics related to natural hazards, disasters, and climate change. Examples include reviews of different natural hazards such as droughts (Woodhouse and Overpeck, 1998), landslides (Aleotti and Chowdhury, 1999), wildfires (Neale and Weir, 2015), tsunami (Chiu and Ho, 2007), and the interactions of those natural hazards (Gill and Malamud, 2014). Others focuses on the impacts (Hunt and Watkiss, 2011) and ecosystem-based adaptation (Brink et al., 2016; Kabisch et al., 2015), education (Johnson et al., 2014), health and psychology after disasters (Kõlves et al.; Harada et al., 2015), volunteerism (Whittaker et al., 2015), disaster management (Beerens and Tehler, 2016; Lettieri et al., 2009; Gall et al., 2015). A significant works on the systematic review of climate change studies has been done by Berrang-Ford et al (2015; 2015; 2012). The author adopts their recommendations for an SLR such as to outline the research questions and aims, data sources and document selection, and analysis and presentation of results. The author conducted a multi-layered literature review to determine inclusion and exclusion for more relevant findings to study publications using the Scopus research engine on publications by February 26th, 2016., with a timeframe from 1900 to 2016. The Scopus research engine was selected because it has the largest database of peer-reviewed literature (Leydesdorff et al., 2010) and has within its features the capability for search, discovery and analysis (Scopus, 2016b).

- In the first stage, the author uses key research terms of *natural hazard, disaster, disaster management, disaster risk reduction, climate change, climate change adaptation, resilience, vulnerability, geology,* and *Indonesia*. The keyword *geology* was added to capture some of the earliest and significant publications on Indonesia which uses the keywords geology and volcanology. This gives 8077 publications.

- The second stage involves exclusions to further refine the results. The exclusion included refinement in subject areas, document types, and source title which did not directly related to the topics. This gives 3447 publications.

- The final stage involves exclusion of those research in the mining industry in Indonesia, those discussed the science of climate change in very general scope and those that touch on the issue of disasters but not specifically in Indonesia. Further exclusions are warranted when the author judges the scope is too broad to be included in the review. The author downloads the results into xml format, saves and imports them into Microsoft Excel. When importing into Excel format the author chooses all delimiters to ensure information went into the right column. However, the results were not always consistent and hence a manual check on each entry row was needed. The author finds that the number counts on the authors´ publications and citations presented in the Scopus search were sometimes different to the actual check of the Excel sheet. Hence, to ensure consistency, higher number of publications and citations are selected. The results in the Excel format are examined line by line to further determine exclusion from the lists. Finally, there are 921 materials selected. The three stages along with the inclusion and exclusion terms are summarized in Table 1.

**Table 1: Multi-stage processes for inclusion and exclusions for search terms**

## 2.2 Data Analysis

The final list was analysed in terms of topics and sub-topics of research citations, keywords, places of focus, types and time of publications, impact factors and authorships. The author used Scopus features to analyse search results such as the article metric module, citation overview, and author profile pages (Scopus, 2016b). The progress of Indonesian scholars is evaluated through counting total number of authors, research outputs and citations overall, and comparing between papers first authored by Indonesians. The author cross-checks the number of citations from Scopus on the Internet through Google and selects the higher citation counts. This was done because it is generally the case that data from a Google search for a publication and author leads to a higher and more up to date citation count. The author also consults total citations and publications of researchers in Google Scholar (Google, 2016c), Research Gate (2016)or from other websites to make sure that the full list of publications are captured. The author checks the organizations, nationalities and genders of the researchers using Google search. There are also cases where the author goes back to Scopus and find author's works to make sure that all are captured.

## 3 Findings and Discussions

This section is structured into two main parts reflecting the objectives of the paper, first on progress of research in terms of key research topics, and second on roles of Indonesian researchers and organizations.

## 3.1 Research Timelines and Topics

The author categorizes the final list into three groups (Table 2), natural hazard, risk, disaster assessments (HRD), disaster risk management and reduction (DRR), and climate change vulnerability, impacts and adaptation (CC), to show and outline how changes in directions on research have taken place over the years and to reduce unbalance towards findings on hazard and risks assessments toward earthquake and volcanic eruption research. In general, there are more research on the topic of HRD (56%), followed by those in DRR (23%), and then CC (21%).

**Table 2: Major Research Topics, descriptions and numbers of publications**

The paper further identifies key periods and timelines by which publications were published. Although the search timeline was set between 1900 and 2016, the years in which publications were found ranges from 1934 to 2016 (Figure 1).

**Figure 1: Number of Publications over the Year**

The first period is from the 1934-1990s. There were no significant changes in the numbers of publications produced. The publications on the HRD are some of the earliest publications indexed in Scopus. It heavily focused on the topics of geophysical hazards and risks related to earthquakes and volcanic eruptions. Within this period, 22 out of 58 events recorded by EMDAT were earthquakes and volcanic activities (EMDAT, 2016). The Bali earthquakes occurred in 1976 and 1979, which in total caused 1764 deaths, affected more than 560 thousands people, and caused more than 200 thousands USD in damages (EMDAT,

2016). The year 1979 was also the year in which the earthquake occurred the most (6 times), in Bali, Lombok, and Biak (near Papua) (USGS, 2016). The second period from the 1990s to 2000s shows a notable increase in the literature, up to an average there 10 publications per year. This gradual increase mainly corresponds to a rise in literature related to the assessments of HRD and is followed by a sharp increase in literature to its highest point in 2000. The third period from 2000s-2016 was the

most dynamic period for publications. While there was a sharp decline since it first peak in 2000, a surge of publications begun in 2004 in response to the Indian Ocean tsunami which especially devastated Indonesia. This increase has continued ever since. This is also a period characterized not only publications related to understanding the risks of earthquakes and tsunami, but also those related to DRR and CC. A peak occurs between 2010 and 2016 which shows soaring published materials in all topics. There were 153 publications in 2016 which is the highest ever produced in a single year. During this period, publications

related to CC has started to be published. Both publications on HRD and CC are expected to rise.

The following sub-sections outline research issues discussed within the three topic groups. Within each, the paper discusses timelines, focus areas of the research, early contributors, and categorization of key topics discussed.

### 3.1.1 Natural hazards, risks and disasters assessments (HRD)

The first sub-section explains findings on the topic of hazards, risks and disasters assessments and identifications. The

EMDAT-CRED (2016) categorization of HRD that is used in this study to help more detailed analysis related to major research topics. Natural-disaster groups only caused by geophysical, meteorological, hydrological, and climatological hazards are included since it is determined that these are the most frequent and impactful disasters in the country.

There are 517 publications in this category. The findings show that there has been a gradual increase in the number of published materials from 1934 to 2000. It first reached its first peak in 2000 that the research in this topic reached its first significant

outputs of 25 publications and reduced slightly after that. In 2004 the Indian Ocean tsunami occurred, initiated with the 9.8 M earthquake with the epicentre off the island of Sumatra, badly affecting Indonesian. Publications related to the tsunami continued to be published until it reached a peak in 2006. Then in 2009, the publications have increased rapidly ever since, reaching another peak in 2016 of 153 publications in a single year. The publications are mostly related to volcanic eruptions, earthquakes and tsunami and the islands of Java and Sumatera are the two areas which receive most attention (more than 70%).

Publications that are related to volcanic eruptions in Java (almost half) such as Merapi (Verstappen, 1988; Lavigne, 1999; Voight et al., 2000; Andreastuti et al., 2000; Charbonnier and Gertisser, 2008; Gertisser et al., 2012; Suryo and Clarke, 1985), Galunggung (Suryo and Clarke, 1985), Semeru (Siswowidjoyo et al., 1997; Carn, 1999; Thouret et al., 2007; Solikhin et al., 2012), Kelud (Lubis, 2014; Nakada et al., 2016) or Ijen (Heikens et al., 2005; Trunk and Bernard, 2008; van Hinsberg et al., 2010). The other hazard that receives many studies is related to the examination of earthquakes (more than 30%), how they

happened, and methods to assess the impacts. The research on tsunami received gradual attention especially after 2004 (Nakamura, 1980; Nakamura, 1978; Latter, 1981; Koshimura et al., 2009; Imamura et al., 1995). There are also a small number of publications related to landslides (Fathani et al., 2016; Karnawati et al., 2011; Liao et al., 2010). Other hazards discussed include those on flood, strong winds, El-Nino, etc (Figure 2).

**Figure 2: Key Topics in HRD Category (Source: modified from SCOPUS results)**

The above findings show that there have been enormous progress publications on this topic. Some of the earliest publications overall also focusses on the characteristics of geophysical hazards and risks. What is needed to be done Most of publication in this topic is however still focuses a lot on geophysical hazards since Indonesia houses some of the most active volcanoes that lies along the "Pacific Ring of Fires" in the world and is located along the fault line of Asian and Australian lines. Studies on the characteristics of earthquakes in terms of hazard assessments are available. What is needed is those related to earthquake risk assessments at the national and smaller scale. The National Agency for Disaster Management (BNPB) has recently developed InaRISK, a web-based service of risk assessments from different hazards (BNPB, 2017). It is however not clear how this information has been utilized for research and most importantly government decision making. The more recent trend of examining hydroclimatic hazards, of floods, landslides, typhoons is encouraging but still not enough. It is quite surprising that studies on flood hazard and risks assessments are still very limited considering that flood is the most frequent disaster and affected the most people in Indonesia (EMDAT, 2017). Most of studies on floods focusses on the impacts on the societies and how government agencies dealt with the impacts. Considering that the impacts of climate-related disasters are increasingly felt in Indonesia, more hazard and risks assessments on floods, typhoons, wildfires, el-Nino are needed, particularly those that examine trends in the past and project future trends.

### 3.1.2 Disaster risk management and reduction (DRR)

The second sub-section is on the topic of disasters risk reduction (DRR). In this study, DRR included those strategies that are aimed at reducing disaster risks and range from risk management to risk reduction including disaster preparedness activities. There are 210 publications in this category. There have been very few publications published before 2003. It is only after 2004 that there was a gradual increase of publications. This reached its peak in 2008, after which the number slightly reduced, before continuing to increase. More than half of the DRR publications focus on Sumatera and Java. However, there are also studies that examine Indonesia as part of worldwide, regional or national assessments.

The topic that receive most attention in this category is related to the governance of DRR (Bakkour et al., 2015; Chang Seng, 2013; Djalante et al., 2013; Djalante et al., 2012; Guarnacci, 2012; Lassa, 2013). The Indonesian government and other stakeholders are actively contributing for DRR (Chang Seng, 2013; Djalante et al., 2013; Djalante et al., 2012; Lassa, 2013). The next key topic is on the evaluation of recovery and reconstruction that have taken place after the 2004 Indian Ocean tsunami (Chang et al., 2011; Daly and Brassard, 2011; Godavitarne et al., 2006; Guarnacci, 2012; Karan and Subbiah, 2011; Telford and Cosgrave, 2007; Lassa, 2015). Other topics that are also related to the impacts of tsunami and disasters were the role of culture, gender , or religion  in helping community resilience when facing disasters, and impacts of disasters on different community groups including children and woman (Baumann, 2008; Donovan, 2010; Donovan et al., 2012; Gaillard et al., 2008b; Islam and Lim, 2015; Balgos et al., 2012; Guarnacci and Di Girolamo, 2012; Hiwasaki et al., 2015; Siagian et al., 2014; Sagala et al., 2009; Schlehe, 2010). Some topics were related to examination of tsunami early warning system (Schlurmann and Siebert, 2011; Steinmetz et al., 2010). There are also many publications which examine the role of knowledge and

information to help communities be more prepared for disasters (Dicky et al., 2015; Hiwasaki et al., 2015; Rafliana, 2012). There are 13 publications comparing Indonesia and Sri Lanka in regards the impacts of the tsunami on how it either become the precursor for peace process in Indonesia but still take time for the process in Sri Lanka (Enia, 2008; Gaillard et al., 2008a; Hyndman, 2009; Kelman, 2005). Some lower numbers of papers examine community-based DRR which is strongly related to community preparedness (Adiyoso and Kanegae, 2013; Birkmann et al., 2015; Hidayati, 2012; James, 2008; Kusumasari and Alam, 2012), and others examine how children are affected psychologically from continuous exposures to hazards and disasters (Du et al., 2012; Lawler and Patel, 2012; Taylor and Peace, 2015; Vignato, 2012), and on emergency management at the local or national level (Esteban et al., 2013; Kusumasari and Alam, 2012; Djalante et al., 2012). Figure 3 summarizes the key topics in DRR category.

**Figure 3: Key topics in DRR category**

The above findings show an encouraging sign on the great variety of research topics related to DRR. This also show a promising sign of development and utilization of social science in understanding the impacts of disasters on the society. The author expects enormous development in this topic. This is also where scholars from Indonesia can contribute significantly. Indonesian scholars have most likely lived in Indonesia for a considerable amount of time. They have experienced, assessed and examined those social and environmental changes that have shaped natural hazards and disasters in the first place, which will help them to be more focused and sharp in terms of documenting. It is however very few studies that examine the legal and regulatory implications of disasters on the government planning, program implementation and the society. While there are organizational reports discussing this (e.g. IFRC, 2016), scholarly articles are rare.

**3.1.3 Climate change risks, vulnerability, impacts and adaptation (CC)**

The third sub-section is related to climate change risks, vulnerability, impacts and adaptation (CC). The research on climate change is interpreted broadly in this paper. The author included all materials that discuss the impacts of climate change not only on disasters caused by natural hazards but also in different sectors such as agriculture, forestry, water and health. This has been done since the current Sendai Framework for DRR calls for multi-risks perspectives and better integration of DRR and Climate change adaptation (CCA) (UNISDR, 2015). There are 194 publications in this category. There have only been a few publications within the period between 1978 and 1990. The second period between 1990 and 2000 saw a slight increase in the literature, and then there were 5 pieces published in 2001.These are related to examinations of the causes and impacts of forest fires in Indonesia. The numbers of publications did not change in general until 2008. It is only after 2010 that there was a sharp increase in the numbers of publications, reaching its peak in 2015 at 35 papers. The islands of Sumatera and Java has become the two major locations for the research of the climate impacts since they are the areas where the greatest number of paddy fields and crops production is concentrated (McCulloch and Peter Timmer, 2008). There are also increasing research related to climate change impacts on different sectors at various locations in Indonesia such as those in Sulawesi and in the eastern part of Indonesia. The author categorizes the publications in this group into three major discussions on impacts of climate change (almost 60%), the governance of CCA (less than 25%), and deforestation and land degradation.

Since most of materials published in this category are related to the review of the impacts on climate change in Indonesia, this paper takes a deeper on those literatures (Figure 4). The impact on crop production, particularly rice, has been the subject of the majority of climate impact researches (Caruso et al., 2016; D'Arrigo et al., 2011; D'Arrigo and Wilson, 2008; Kawanishi and Mimura, 2015; Keil et al., 2009; Naylor et al., 2001; Sano et al., 2013; Shofiyati et al., 2014) which is strongly associated with droughts (Aldrian and Djamil, 2008; D'Arrigo and Smerdon, 2008; D'Arrigo and Wilson, 2008; D'Arrigo et al., 2006; Keil et al., 2009; Keil et al., 2008). A high number of publications link droughts (Salafsky, 1994; D'Arrigo et al., 2006; D'Arrigo and Smerdon, 2008; Shofiyati et al., 2014) and forest fire (Usman and Hartono, 1997; Fang and Huang, 1998; Brauer and Hisham-Hashim, 1998; Jim, 1999; Stolle and Tomich, 1999; Page et al., 2002; Stolle and Lambin, 2003). Studies on water are related to impacts of climate change on ocean circulation (Susanto, 2001), water availability and quality (Rai et al 2015), and management (Poerbandono et al, 2014), especially those in urban area (Larson et al, 2013) and major river basins (Sahu et al., 2012). Floods and sea level rise are another topic received strong interest (Marfai and King, 2008; Marfai et al., 2008; Marfai et al., 2015; Muis et al., 2015; Neolaka, 2013, 2012; Sarminingsih et al., 2014; Shrestha et al., 2014), particularly on the impacts on coastal communities and cities (Budiyono et al., 2016; Ward et al., 2013; Firman et al., 2011; Wassmann et al., 2009; Nicholls et al., 1995). The impact of climate change on health is on tropical diseases (Coughlan de Perez et al., 2015) and impacts of increased temperatures on animal (Purnomo et al., 2011; Morwood et al., 2008). Indonesia houses some of the largest of rainforest, in Sumatera and Kalimantan. Forestry issues is discussed in relation to reducing emissions from deforestation and forest degradation, forest conservation and sustainable management, and enhancement of forest carbon stocks (REDD+) (Cerbu, Swallow and Thompson, 2011; Saatchi *et al.*, 2011; Baccini *et al.*, 2012; Margono *et al.*, 2012; Hansen *et al.*, 2013; Minang *et al.*, 2014). A small number of research are on the changing pattern of temperature and rainfall (D'Arrigo and Wilson, 2008; Aldrian and Djamil, 2008; Chrastansky and Rotstayn, 2012).

**Figure 4: Key Topics in CC Category Researching on Impacts of Climate Change**

The above findings show that research on CC has the least progress amongst the other topics. This is an outmost concern considering that Indonesia is one of the most vulnerable countries to climate change (UNU-EHS, 2015). It is however encouraging to see that the range of research in this topic varies in terms of impacts on agriculture, water, health, and forestry sectors. Indonesia is the third largest emitter of Greenhouse Gasses Emissions especially from deforestation and the situation is reflected in the literature. It is imperative that more studies are needed to understand the vulnerability of the society to climate change, especially since 80% of its population lives along the low lying coastal areas (Neumann et al 2015). Future societal disruptions due to probable loss of livelihoods, environmental migration, climate-induced conflicts needed to be understood. It is also important to equip decision makers on how to deal with climate impacts through mainstreaming in development planning.

**3.2 Progress and Roles of Indonesian Researchers and Organizations**

This second section examines the roles of Indonesian researchers and organizations in contributing to the production of literature. It first describes some of the earliest literatures on each category. It also addresses to what extent Indonesian

researchers have been collaborating with other international/non-Indonesian researchers and organizations, and in producing high impact English journal articles. The roles of authors are examined in general term, and specifically looking at the 10 highly cited papers with Indonesian as first author.

### 3.2.1 Authorships

5 The oldest publications listed in Scopus are those by Reinout Willem van Bemmelen, a Dutch national born in Batavia (Netherlands East Indies/Indonesia), on *Ein Beispiel für Sekundärtektogenese auf Java* (An example of secondary isogenesis on Java) (van Bemmelen, 1934) and *Über die Deutung der Schwerkraftanomalien in Niederländisch-Indien* (On the Interpretation of the Gravity Anomalies in Dutch-India) (van Bemmelen, 1935), both from the *Geologische Rundschau* (now listed as the International Journal of Earth Sciences). Van Bemmelen continued to write extensively on theories in

10 Techtonophysics, and on Indonesia (van Bemmelen, 1935, 1941, 1949b, 1953, 1963). He then wrote in English on the Origin and Mining of Bauxite in Netherlands-India (Van Bemmelen, 1941) and on the Report of Volcanic Activity and Vulcanological Research in Indonesia (1936-1948) (van Bemmelen, 1949b) in the *Bulletin of Volcanologique*. These works formed his most significant contribution: *The Geology of Indonesia* (Van Bemmelen, 1949a; Van Bemmelen and Bourter, 1970). In addition, Rittman (1953) wrote specifically on the Magmatic Character and Tectonic Position of Indonesian Volcanoes. In terms of

15 contributions by Indonesian researchers, John Ario Katili of the Bandung Institute of Technology (ITB), considered one the founding fathers of Indonesian Geology, wrote significant accounts on geotectonic knowledge of Indonesia from the period of 1963 to 1991 (Katili, 1975, 1991, 1974, 1967, 1971, 1989, 1969a, 1978, 1986, 1981b; 1963; 1969b, 1981a, 1980, 1973). Other early and significant contributions come from Mudaham Taufick Zen and Djajadi Hadikusumo, from the Geological Survey of Indonesia, who collaboratively wrote some of the earliest and most important accounts on volcanoes in Indonesia

20 (Zen and Hadikusumo, 1965, 1964b, a; 1971, 1970, 1966; 1974). It is also important to mention, though not indexed in Scopus, the work by Kusumadinata (1979), of the Geological Survey of Indonesia, on the *Catalogue of References on Indonesian Volcanoes with Eruptions in Historical Time*, amongst others (Kusumadinata, 1963, 1964a, b, c; cited in Rampino and Self, 1982).

The earliest accounts that explicitly examine DRR include Suryo and Clarke (1985) who wrote on the Occurrence and

25 Mitigation of Volcanic Hazards in Indonesia, and laid out strategies such as the prediction of volcanic activity, hazard zoning and maps, and control of hazards through engineering structures. They wrote that ´*the main purpose of hazard maps is to assist the protection of people and their property near active volcanoes*´ (Suryo and Clarke, 1985, p. 90). Verstappen (1994; 1993, p. 367) in his paper, the Volcanoes of Indonesia and Natural Disaster Reduction (with Some Examples), wrote that ´*since emergency scenarios inevitably vary with intensity and type of land utilization, the compilation of vulnerability maps of the*

30 *endangered areas merits consideration in the context of disaster reduction policy*'. An Indonesian notable scholar is Sudibyakto, from the Faculty of Geography, University of Gadjah Mada, and also the head of the Indonesia Disaster Scientist Association (IABI), who wrote Natural Disaster Mitigation and Management in Indonesia (Sudibyakto and Haroonah, 1997) and examine disaster from geographical and social science perspectives (Sudibyakto and Haroonah, 1997; 1992; 1996).

Some earliest publication were written in 1992 by Sudibyakto (1992) who wrote *Facts and Future Trends of Climate Change: A Case Study of the Eastern Part of the Indonesia Islands*, and by Murdiyarso (1993) who examined the management of climate change impacts to reduce $CO_2$ release resulting from deforestation and biomass in Indonesia.

Figure 5 summarizes the roles of Indonesian authors within each publication category (HRD, DRR, and CC). The review finds that out of the 3,000 names obtained from the Scopus search, there are 68% of international authors compared to 32% Indonesian author. The contribution of international/non-Indonesian authors dominates the production of publications. The figure shows that there are more authors, including Indonesian authors, in DRR category than the other two categories. There are slightly more papers with at least one Indonesian author than those with no Indonesian authors. A more striking examination of Indonesia authors shows that there are less than 100 authors with more than 2 publications. The majority of authors work for organizations that are located in Java where the high quality education providers are mostly located (OECD and ADB, 2015), dominated by male researchers and only a small minority of these researchers have social media account such as Google Scholar (Google, 2016a) or Research Gate (2016b) or professional and personal websites. This implies that there is room for increasing the involvement of Indonesian authors writing about various issues related to DRR, and a greater opportunity for developing social science in DRR. More Indonesians need to be involved in international publications and specific interventions are needed to enhance writing, publication and outreach skills.

**Figure 5: Comparing the roles of international and Indonesian authors in each publication category**

Table 3 compares the list of the top ten authors with highest number of publications and the Indonesian authors with the 10 highest publications. Highest in the list is Hasanuddin Zainal Abidin of the Bandung Institute of Technology (ITB), with 71 publications listed in Scopus, while his Google scholar profile shows that he has published 172, with 1709 citations (Google Scholar, 2016b). Franck Lavigne from *Université* Paris 1 Pantheon Sorbonne published the second highest numbers of papers (Google Scholar, 2016a). Lavigne worked closely with Jean-Claude Thouret from *Laboratory Magmas et Volcanis* (LMV, 2016). Danny Hilman Natawidjaja works for Indonesian Institute of Science (LIPI) (Google Scholar, 2016c) but did his bachelor study from Bandung Institute of Technology (ITB). Kerry Sieh, from Earth Observatory of Singapore (EOS), has long collaborated with Natawidjaja on their works on seismology in Indonesia (EOS, 2016). Barry Voight is a renowned geologist and volcanologist in USA who has worked on the Mount Merapi since the 1980s (Google Scholar, 2016e). Ralf Gertisser is a senior lecturer in Keele University (Google Scholar, 2016d). Bambang Widoyoko Suwargadi is affiliated with LIPI and Surono (1 name only) and Muhammad Hendrasto both work for the Center for Volcanology and Geological Hazard Mitigation (PVMBG, 2016). In addition to the 5 Indonesians in the top 10 authors, Irwan Meilano, Heri Andreas and Irwan Gumilar have worked closely with Abidin and are all affiliated with ITB. Muh Aris Marfai and Junun Sartohadi are from the Gadjah Mada University (UGM). This result shows a great deal of need for increasing the capacity of Indonesian authors meet standards for internationally regarded journal publications. There are a limited number of authors involved with publications in the highest impact factor (IF) journals such as Nature and Science. Indonesian authors largely lack experience in international collaboration and the language and writing skills necessary for submitting their works to internationally accredited journals: High impact articles and collaborations were only done through organizations centred on ITB, UGM, LIPI

and PVMBG. Despite some Indonesian researchers who have been strongly influential within the study of hazards, DRR or climate change in Indonesia and could potentially contribute to the global development of knowledge in these fields, they have only published in Bahasa Indonesia and did not submit their works into international mostly English language journals.

**Table 3: List of top ten authors with highest number of publications, and top ten Indonesian authors (SCOPUS, 2016a; Google, 2016b; Research Gate, 2016a)**

### 3.2.2 Affiliations

This section systematically examines the place, from regional to national, and organizations by which the researchers are affiliated in Indonesia. The organizations which house the ten most productive publications related to this review are shown in Figure 6. In general, there are an equal number of organizations that are based in Indonesia, and their contributions comprised slightly more than half the overall contributions amongst these most productive agencies. This paper looks deeper at the contribution of different organizations within Indonesia. It is shown that ITB and UGM dominate almost half the total publications. There are also more twice universities in Java that those outside Java, while the rest of publications are contributed by national level organizations such as LIPI and PVMBG.

**Figure 6: Organizations with highest number of publications (Indonesian Organizations marked with \*)**

### 3.2.3 Publications Sources

This section presents the source of publications. Most of publications from journals are those that got indexed, compared to conference proceedings, books, or others. A closer look at the journals shows those related to geophysical hazards (volcanoes, earthquakes, tsunami, etc) identification and assessments dominate the numbers of papers published on Indonesia (Table 4). Moreover, the Indonesian Journal of Geography is the only Indonesian journal that is found this review. The journal was established in 1961 by the Faculty of Geography, UGM in cooperation with the Association of Indonesian Geographers (UGM, 2016). There are no clear counts on the number of academic journals in Indonesia, however, there are only 245 accredited by DIKTI (Higher education directorates of the Ministry of Education) (DIKTI, 2016b) and 17 indexed in SCOPUS (DIKTI, 2016a). In addition, none of these journals have yet obtained an impact factor, and hence a Scientific Journal Ranking (SJR) Score is presented instead (SJR, 2016).

**Table 4: List of most submitted journals**

### 3.2.4 Citations

This section analyzes the citations for each topic category. Overall, the HRD category has the highest number of citations, in total more than two thirds (3945/5291) of all citations. A look of the citation averages, however, shows quite a different story. Whilst the CC literature category has the least number of papers published (194), the citation average is twice of the DRR category (3,18). Figure 7 shows the comparison between the progress of Indonesian researchers in the 10 most cited papers overall and those first authored by Indonesians. The role of first author has been considered significant since they are traditionally assumed to lead the research and write most of the content, and therefore receive most credit (Riesenberg and

Lundberg, 1990; Hu, 2009). It shows that there are more authors, mostly international authors in the 10 most cited papers, while there are more Indonesians in the 10 most cited papers first authored by Indonesians. This might suggest that Indonesian researchers tend to work with other Indonesians and hence needed to expand their collaborations with international scholars as a strategy to increase their number of citations and ability to submit for higher impact journals.

**Figure 7: Comparing the roles of Indonesian researchers in the 10 most cited papers**

Table 5 shows the list of the 10 most cited papers of all publications. Within the 10 most cited papers, the total citations are 4,204 with a combined impact factor (IF) of 293.618, and only one third of the authors are Indonesian. The citation is three times of those first authored by Indonesians, and the IF is 4 times greater. It is shown that they are published in high impact factor journals such as Nature, Science, or those related to geophysical hazards. The two highest cited papers are published in

Nature and discuss the impacts of forest fires in Indonesia. The paper related to the examination of the amount of carbon released from peat and forest fires in Indonesia in 1997 has the highest citation of 1287 by Page et al (2002). The majority of the papers discuss major hazards from the earthquake in Sumatera (Ishii et al., 2005; Briggs et al., 2006; Hsu et al., 2006; Konca et al., 2008), to the impacts of Toba (Rampino and Self, 1992) and Merapi volcanic eruptions (Voight et al., 2000). Eight papers were also contributed by Indonesians with Natawidjaja was involved in five of them. Adi Jaya and Suwido Limin

are both lecturers from the Palangkaraya University in Kalimantan, where forest fires frequently occurred across the rain forest and impacted not only Indonesia but also surrounding countries in the region such as Singapore (Tay, 1998) and Malaysia (Khandekar et al., 2000). Natawidjaja and Subarya, along with Sieh contributed the most (Briggs et al., 2006; Hill et al., 2012; Horspool et al., 2014; Hsu et al., 2006; Konca et al., 2008; Muhari et al., 2010; Nalbant et al., 2005; Philibosian et al., 2012; Prayoedhie et al., 2012; Schlurmann et al., 2010; Singh et al., 2010).

A closer examination of the list of ten most cited publications with Indonesian first authors shows a very striking picture. The total citations are only 1542, with a combined IF of only 70, 012, with 80% of all authors being Indonesian. The papers are much more varied in terms of topics they discussed. The first two most cited papers are related to impacts of climate change in Indonesia. Aldrian (2003), Susanto (2003; 2001) and also Amien et al (1996) authored papers related to climate change or its impacts on Indonesia. Natawidjaja (Natawidjaja et al., 2006; Natawidjaja et al., 2004) and Abidin (Abidin et al., 2001;

Abidin et al., 2011) both have 2 papers to contribute each within the list of most cited papers first authored by Indonesian on earthquakes and land subsidence assessments. One paper examines the impacts of volcanoes (Andreastuti et al., 2000). Marfai wrote extensively on coastal risks and disasters in cities such as Semarang or Jakarta (Marfai and King, 2008; Marfai et al., 2008; Marfai et al., 2015; Ward et al., 2013; Marfai, 2014; Marfai and King, 2007). This table shows that in generals, Indonesia authors still write papers with fewer citations, and the organizations that house these authors are still extremely limited to ITB,

UGM, LIPI, and PVMBG. Another significant finding here is that there is no paper on DRR. This is an important finding that which also show how social science perspectives needed to be taken up by the Indonesia researchers in dealing with the management of disaster risks and disaster risks in Indonesia.

**Table 5: Comparing Citations Authored in General and Those First Authored by Indonesian in 10 Most Cited Papers**

## 4 Recommendations for Future Research and Policy Relevance, and Conclusions

This paper has presented the results of a systematic literature review from Scopus to on the current research trends and progress related to natural hazards, disasters, and disaster risks reduction, as well as increasingly climate change impacts and governance in Indonesia. The paper also examines the roles of Indonesian authors and organizations in contributing to publications related to these topics. We have seen that some of the earliest publications were written in 1934 and publications started to increase rapidly since 2000. It is found there are more publications on HRD, than those on DRR and CC. Moreover, there are twice international authors for every Indonesian author and the contribution of international authors dominates the production of publications. Male and advanced career authors still dominate, compare to the numbers and roles of female and early career researchers (ECR). Most of the high impact publications and international collaborations were conducted with the key institutions centred on ITB, UGM, LIPI and PVMBG. In addition, there are very few researchers have social media accounts such as Google Scholar (Google, 2016a) or Research Gate (Research Gate, 2016b) or professional and personal websites.

The *first* recommendation is related to future research topics. More research is needed on different hazards, different locations in Indonesia, and other topics in DRR and climate change. Majority of current research is still focused on geophysical hazards and those related to hydro-meteorological hazards have only received attention recently. It has been seen that majority of research focuses on the Islands of Java and Sumatera. This is expected since both islands are the most at risks from natural hazards in Indonesia. Multi hazard, risks and vulnerability assessments are suggested. Research and actions that focus on the most vulnerable places and communities are needed.

As the world is increasingly urbanized, there is strong international attention focusing and reducing risks in urban areas, in particular through concerted action in the New Urban Agenda (UN HABITAT, 2016). More research need to consider the context of urban areas by which social risks and risks from natural hazards play out simultaneously, and the impacts on urban dwellers needs to be understood. Cities in Indonesia like Jakarta, Surabaya or Makassar are rapidly urbanizing and environmental and economic pressures increase risks for the inhabitants (Firman et al., 2011; Larson et al., 2013; Santosa, 2000; Firman, 2016; van Voorst, 2016).

Strategies and actions for integrating DRR and CCA need to be explored further (Djalante and Thomalla, 2012; Lassa and Nugraha, 2015). Disaster risk governance has not received much research especially on the interplay with decentralization which places responsibility for DRR and CCA at the local government level (Lassa, 2013; Kusumasari et al., 2010). The strategies outlined are not only relevant for research but also for the governance for climate change. The islands in Kalimantan, Sulawesi, Maluku and Papua in the eastern part of Indonesia have also been impacted by droughts, floods or strong winds and needs to be addressed in the future. The impacts of sea level rise on small islands, drought on forests in Kalimantan and Papua, raising sea level and ocean acidification on fisheries industry in Sulawesi and eastern part of Indonesia, are some of the increasingly worrisome issues expected from climate change. There is still greater need for research and government actions on climate change topics related to linkages between poverty and disaster vulnerability (Suryahadi and Sumarto, 2003), security (CSIS, 2016), loss and damages (Warner et al., 2012), impacts on key sectors such as fisheries (USAID Indonesia,

2015), coastal communities (Marfai, 2014; Marfai et al., 2008), food security (Measey, 2012; WFP, 2015) health (Ady Wirawan, 2010; Haryanto, 2009), migrations (Raleigh et al., 2008; Reuveny, 2007), and community-based DRR (Heijmans, 2012). Many activities done by the Indonesian government and international and development agencies on their implementations for DRR or CCA programmes have focused at different administrative level from national, regional, local

and on the community level. There is abundance of activity reports by governments, donor and international agencies (e.g. USAID, 2016; USAID Indonesia, 2011, 2015); however, those reports are rarely made available or submitted for academic publications.

The *second* recommendation is on the need to strengthen the capacity of research collaborations between Indonesian and international researchers, multi-disciplinary research and publications in high impacts journals, along with the need for

strengthening of science communication to social media outlets and science-policy advocacy. There needs to be more funding and incentives for collaborations. More trainings on English academic writing and journal article publications are needed, including capacity building for early career, female and social science researchers. It is clear that some of the very limited Indonesian research from key universities doing disaster research such as ITB, LIPI, UGM have been involved in international collaborations and publications of high impacts journal (QS, 2016). There are only nine universities  in Indonesia that are

within the list of QS World University Rankings, with University of Indonesia at the top of the list (QS, 2016). Other universities on the islands of Sumatra, Sulawesi, and Kalimantan and other locations need to address disaster issues as part of their research agendas (OECD and ADB, 2015). There is a need for better targeting of scholars to do more collaboration for research and writing for high impact journals. This goes along with strengthening the capacity of researchers and lecturers at the universities to write and publish in international journals. The Ministry of Education has indeed conducted a training

scheme and provided incentives for lecturers that have published internationally (RISTEKDIKTI, 2016), however, the overall quality and quantity of papers by Indonesian researchers are still much less that those at comparable universities in Malaysia or Singapore (RISTEKDIKTI, 2016). The list from Scopus shows that there is still only small numbers of female and early career researchers (SCOPUS, 2016a). The first stage is to have proper identification of researchers and make this available to public. The author could not find a repository of researchers from the Ministry of Education website, let alone systematically

determining their progress, history of schooling and research. There have been some concerns to strengthen the capacity of female researchers globally (Larivière et al., 2013), and also similarly in Indonesia. Early career researchers (ECR) are defined as those who are within 8 years after completing PhDs or within 6 years of trainings (AHRC, 2016). While globally there has been some systematic efforts to strengthen the capacity of ECR such as through mentoring (Clarke, 2004; Kram and Isabella, 1985), there are no clear strategies  for the Indonesian ERC from the Indonesian governments. International journals (Elsevier,

2016) and international and other national research council (RCUK, 2016) have allocated resources and are funding research specifically for ECR. The Indonesian Association of Disaster Experts was formed in 2014 and meets annually to discuss their future research guidelines (IABI, 2016). One thing that should be on the agenda is to review current publications in Bahasa Indonesia and collaborations undertaken by Indonesian experts which can enable better identification of research progress and hence research needs in the future. There is abundance of materials within Indonesian repositories related to *bencana*

(Indonesia word for disaster), especially within the repositories at ITB, UGM, and University of Syiah Kuala in Aceh. These materials and research activities done within the universities need to be reviewed and submitted for international journals to give a broader view on issues that have been discussed by scholars in Indonesia. There is increasing call for more inter-disciplinary collaborations so that complex problems on social and environmental issues can be understood better and problems identifications can better target those in needs (Future Earth, 2016). Hence this implies increasing importance of social science adoption to study disasters and their impacts. The roles of private business and the communities at risk have rarely been part of the research and collaborations. It is also not clear how collaborations amongst scientists from social and physical backgrounds have taken place in Indonesia. It is also not clear how or whether science (Wagner and Leydesdorff, 2005a), policy and industry (Lee, 1996) collaborations have taken place and were documented in these listed publications. These collaborations are important to face the complexities of future problems (Leydesdorff and Wagner, 2008), and also to help achieve the outcomes of the Sustainable Development Goals (United Nations, 2015).

In conclusion this study has been able to determine the progress in research related to natural hazards, risks, and risk reduction and climate change impacts in Indonesia. It has also been able to examine the roles of Indonesian scientists in collaborations and towards more and also high-quality publications. The recommendations are outlined toward these two issues and it is the responsibility of both Indonesian and international organizations including governments that have worked and will work in Indonesia to be able to meet the needs for Indonesia to better understand, manage, and reduce its natural hazards and risks in the future and ultimately build a resilient and sustainable Indonesia.

**Acknowledgment**

The author would like to acknowledge the Alexander von Humboldt Fellowship for Experienced Researchers which facilitates her research visit (August 2015-July 2017) in Germany at the United Nations University Institute for Environment and Human Security, Germany. She would like to thank Dr. Matthias Garschagen as the head of VARMAP section of UNU-EHS for his supports received during her research in Germany. The author benefits enormously from the reviewers' comments and has greatly improved to quality of the paper. As of August 2017, the author is based at the United Nations University Institute for the Advanced Study of Sustainability, Japan.

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

**List of Tables**

**Table 1: Multi-Stage Processes for Inclusion and Exclusions for Search Terms**

| Stage | Inclusion / Exclusion | Description | Search terms | Results |
|---|---|---|---|---|
| First | Inclusion based on Search Terms | Keywords | (TITLE-ABS-KEY(hazard*) OR TITLE-ABS-KEY(risk*) OR TITLE-ABS-KEY(disaster*) OR TITLE-ABS-KEY(disaster management*) OR TITLE-ABS-KEY(disaster risk reduction*) OR TITLE-ABS-KEY(climate change*) OR TITLE-ABS-KEY(climate change adaptation*) OR TITLE-ABS-KEY(resilien*) OR TITLE-ABS-KEY(vulnerabili*) OR TITLE-ABS-KEY(volcan*) OR TITLE-ABS-KEY(geolog*) AND TITLE-ABS-KEY(Indonesia)). | 8077 |
| Second | Exclusion on keywords | Those that are relate to clinical/health studies | AND ( EXCLUDE ( EXACTKEYWORD , "Human" ) OR  EXCLUDE ( EXACTKEYWORD , "Humans" ) OR  EXCLUDE ( EXACTKEYWORD , "Female" ) OR  EXCLUDE ( EXACTKEYWORD , "Male" ) OR  EXCLUDE ( EXACTKEYWORD , "Adult" ) OR  EXCLUDE ( EXACTKEYWORD , "MajorClinicalStudy" ) OR  EXCLUDE ( EXACTKEYW | 3447 |

| Stage | Inclusion / Exclusion | Description | Search terms | Results |
|---|---|---|---|---|
| | | | ORD , "ControlledStudy" ) OR EXCLUDE ( EXACTKEYWORD , "Adolescent" ) OR E XCLUDE ( EXACTKEYWORD , "Prevalence" ) OR EXCLUDE ( EXACTKEYWORD , "Child" ) OR EXCLUDE ( EXACTKEYWORD , "Thailand" ) OR EXCLUDE ( EXACT KEYWORD , "Aged" ) OR EXCLUDE ( EXACTKEYWORD , "China" ) OR EXCLUD E ( EXACTKEYWORD , "India" ) OR EXCLUDE ( EXACTKEYWORD , "Infant" ) OR EXCLUDE ( EXACTKEYWORD , "Developing Country" ) )  OR ( EXCLUDE ( EXACTKEYWORD , "Gold" ) ) | |
| | Exclusion on subject area | Only those in environmental studies in general | AND ( EXCLUDE ( SUBJAREA , "ENER" ) OR EXCLUDE ( SUBJAREA , "MEDI" ) OR EXCLUDE ( SUBJAREA , "BIOC" ) OR EXCLUDE ( SUBJAREA , "CENG" ) OR EXCLUDE ( SUBJAREA , "MATE" ) OR EXCLUDE ( SUBJAREA , "CHEM" ) OR E XCLUDE ( SUBJAREA , "NURS" ) OR EXCLUDE ( SUBJAREA , "DECI" ) OR EXC LUDE ( SUBJAREA , "PHAR" ) OR EXCLUDE ( SUBJAREA , "IMMU" ) OR EXCLU DE ( SUBJAREA , "NEUR" ) OR EXCLUDE ( SUBJAREA , "DENT" ) OR EXCLUDE ( SUBJAREA , "Undefined" ) ) | |
| | Exclusion on tittle | Tittles are deemed unrelated | AND ( EXCLUDE ( EXACTSRCTITLE , "ChemicalGeology" ) OR EXCLUDE ( EXAC TSRCTITLE , "Journal Of Petrology" ) OR EXCLUDE ( EXACTSRCTITLE , "Contributions To Mineralogy And Petrology" ) OR EXCLUDE ( EXACTSRCTITLE , "SPE Asia Pacific Oil And Gas Conference" ) OR EXCLUDE ( EXACTSRCTITLE , "International Conference On Health Safety And Environment In Oil And Gas Exploration And Production" ) OR EXCLUDE ( EXACTSRCTITLE , "Society Of Petroleum Engineers International Petroleum Technology Conference 2012 Iptc 2012" ) OR EXCLUDE ( EXACTSRCTITLE , "Acta Horticulturae" ) OR EXCLUDE ( EXACTSRCTITLE , "Preventive Veterinary Medicine" ) OR EXCLUDE ( EXACTSRCTITLE , "SPE Asia Pacific Oil Gas Conference" ) OR EXCLUDE ( EXACTSRCTITLE , "Society Of Petroleum Engineers SPE Asia Pacific Oil And Gas Conference And Exhibition 2011" ) OR EXCLUDE ( EXACTSRCTITLE , "Australasian Institute Of Mining And Metallurgy Publication Series" ) OR EXCLUDE ( EXACTSRCTITLE , "Australian Systematic Botany" ) OR EXCLUDE ( EXACTSRCTITLE , "Society Of Petroleum Engineers SPE Asia Pacific Oil And Gas Conference And Exhibition Apogce 2013 Maximising The Mature Elevating The Young" ) OR EXCLUDE ( EXACTSRCTITLE , "AAPG Bulletin American Association Of Petroleum Geologists" ) OR EXCLUDE ( EXACTSRCTITLE , "Bird Conservation International" ) OR EXCLUDE ( EXACTSRCTITLE , "Construction And Professional Practices Proceedings Of The 10th East Asia Pacific Conference On Structural Engineering And Construction Easec 2010" ) OR EXCLUDE ( EXACTSRCTITLE , "Corporate Ownership And Control" ) OR EXCLUDE ( EXACTSRCTITLE , "Undefined" ) ) | |
| | Exclusion on language | Those that are not written in English and Bahasa Indonesia are excluded | AND (EXCLUDE (LANGUAGE , "Italian" ) OR EXCLUDE ( LANGUAGE , "Polish" ) OR EXCLUDE ( LANGUAGE , "Spanish" ) OR EXCLUDE ( LANGUAGE , "Afrikaans " ) OR EXCLUDE ( LANGUAGE , "Swedish" ) ) | |
| | Exclusion on subject area | Those that are too broad on the subject area are excluded | AND (EXCLUDE ( SUBJAREA , "ECON" ) OR EXCLUDE ( SUBJAREA , "COMP" ) OR EXCLUDE ( SUBJAREA , "BUSI" ) OR EXCLUDE ( SUBJAREA , "MATH" ) OR EXCLUDE ( SUBJAREA , "PSYC" ) OR EXCLUDE ( SUBJAREA , "VETE" ) OR EX CLUDE ( SUBJAREA , "HEAL" ) ) | |
| | Exclusion on document type | Only journal articles are included | AND (EXCLUDE ( SRCTYPE , "d" ) OR EXCLUDE ( SRCTYPE , "r" ) ) AND ( EXCL UDE ( DOCTYPE , "cr" ) OR EXCLUDE ( DOCTYPE , "no" ) OR EXCLUDE ( DOCT YPE , "sh" ) OR EXCLUDE ( DOCTYPE , "ed" ) ) | |
| Final | Transfer Exclusion | Those that are too broad on the subject area are excluded | Transfer to XML and excel Form Topics that are too broad, e.g. mining, general climate science, minor mention or not directly on Indonesia | 921 |

**Table 2: Major Research Topics, Descriptions and Numbers of Publications**

| Major topics groups | Definitions (IPCC, 2012; UNISDR, 2009) | Number of publications (Percentage) |
|---|---|---|
| (1) Natural hazard, risks, disasters assessments (HRD) | Hazards: A dangerous phenomenon, substance, human activity or condition that may cause loss of life, injury or other health impacts, property damage, loss of livelihoods and services, social and economic disruption, or environmental damage (UNISDR). Risks: The combination of the probability of an event and its negative consequences. Disaster: A serious disruption of the functioning of a community or a society involving widespread human, material, economic or environmental losses and impacts, which exceeds the ability of the affected community or society to cope using its own resources (UNISDR). | 517 (56%) |
| (2) Disaster risk management or reduction (DRR) | The systematic process of using administrative directives, organizations, and operational skills and capacities to implement strategies, policies and improved coping capacities in order to lessen the adverse impacts of hazards and the possibility of disaster (UNISDR). The concept and practice of reducing disaster risks through systematic efforts to analyze and manage the causal factors of disasters, including through reduced exposure to hazards, lessened vulnerability of people and property, wise management of land and the environment, and improved preparedness for adverse events (UNISDR). | 210 (23%) |
| (3) Climate change vulnerability, impacts and adaptation (CC) | Climate Change: A change of climate which is attributed directly or indirectly to human activity that alters the composition of the global atmosphere and which is in addition to natural climate variability observed over comparable time periods (IPCC). Climate change adaptation: The adjustment in natural or human systems in response to actual or expected climatic stimuli or their effects, which moderates harm or exploits beneficial opportunities (UNISDR). | 194 (21%) |
| **Total** | | 921 |

**Table 3: List of top ten authors with highest number of publications, and top ten Indonesian authors (SCOPUS, 2016a; Google, 2016b; Research Gate, 2016a)**

5   Note: NoP =Number of Publications, SC= SCOPUS Profile (publications, citations, h-index, number of co-authors, most frequent collaborator), GS = Google Scholar profile (citations, h-index, i10-index), RG = Research Gate profile (number of publications, citations, impact points), N/A = Not Available

| Top 10 Author (I=Indonesian) | Organization / Country | NoP | SC | GS | RG | Top 10 Indonesian Author | Organization | NoP | SC | GS | RG |
|---|---|---|---|---|---|---|---|---|---|---|---|
| Abidin, HZ (I) | Indonesia / Institute Teknologi Bandung (ITB) | 71 | 71, 571, 11, 150, Andreas, H | 172, 1709, 41 | 119, 773, 99.21 | Abidin, HZ | ITB | 71 | 71, 493, 11, 121, Andreas H | N/A | 119, 773, 99.21 |
| Lavigne, F | France / Université Paris 1 Panthéon Sorbonne | 59 | 66, 1356, 20, >150, Wassmer, P | 124, 1648, 34 | 153, 1430, 162.61 | Meilano, Irwan | ITB | 47 | 46, 299, 10, 143, Kimata, F | 514, 11, 14 | 24, 69, |

| Top 10 Author (I=Indonesian) | Organization / Country | NoP | SC | GS | RG | Top 10 Indonesian Author | Organization | NoP | SC | GS | RG |
|---|---|---|---|---|---|---|---|---|---|---|---|
| Sieh, K | Singapore / Earth Observatory of Singapore | 54 | 120, 5752, 43, >150, Natawidjaja, DH | N/A | N/A | Natawidjaja, DH | (Indonesian Institute of Science) LIPI | 43 | 43, 1913, 21, 123, Sieh KE | 147, 2964, 25, 33 | 123, 2788, 376.31 |
| Natawidjaja, DH (I) | Indonesia / LIPI | 43 | 42, 1913, 21, 123, 33 Sieh KE | 147, 2964, 25, 33 | 123, 2788, 376.31 | Suwargadi, BW (I) | Indonesia / LIPI | 31 | 31, 1102, 17, 103, Natawidjaja, DH | 97, 1585, 20, 24 | N/A |
| Thouret, J-C | France / Laboratory Magmas er Volcanis | 40 | 114, 1147, 20, >150, Gourgaud, A | N/A | N/A | Surono (1 name only) | (Center for Volcanology and Geological Hazard Mitigation) PVMBG | 28 | 28, 348, 12, 125, Hendrasto M | N/A | N/A |
| Voight, B | USA / Pennsylvania State University | 36 | 313, 8185, 53, 128 | 2505, 307 570, 75 | | Andreas, H | ITB | 24 | 24, 123, 6, 46, Abidin, H Z | N/A | N/A |
| Gertisser, R | United Kingdom / Keele University | 32 | 42, 684, 468, 14, >150, Charbonnier SJ | 86, 1009, 19, 29 | 87 803 132, 51 | Marfai, MA | Gadjah Mada University (UGM) | 21 | 183, 8, 36, King, L | 79, 517, 12, 14 | N/A |
| Suwargadi, BW (I) | Indonesia / LIPI | 31 | 31, 1102, 17, 103, Natawidjaja, DH | 97, 1585, 20, 24 | N/A | Gumilar, I | ITB | 20 | 20, 68, 3, 44, Abidin HZ | N/A | N/A |
| Surono (I) | Indonesia / PVMBG | 28 | 28, 448, 13, 129, Hendrasto M | N/A | N/A | Sartohadi, J | UGM | 19 | 19, 378, 8, Lavigne, F | N/A | N/A |
| Andreas, H (I) | ITB | 24 | 123, 6, 46, Abidin, H Z | N/A | N/A | Hendrasto M | PVMBG | 18 | 18, 92, 4, Surono | N/A | N/A |
| Total | | 416 | | | | | | 306 | | | |

**Table 4: List of most submitted journals (source: modified from SCOPUS results)**

| Publications | Number of papers | IF / SJR | Category | | |
|---|---|---|---|---|---|
| | | | HRD | DRR | CC |
| 1. Journal of Volcanology and Geothermal Research | 75 | 2.543 | x | | |
| 2. Natural Hazards | 39 | 1.719 | x | x | |
| 3. Natural Hazards and Earth System Science | 27 | 1.735 | x | x | |
| 4. Bulletin of Volcanology | 22 | 2.519 | x | | |
| 5. Geophysical Research Letters | 17 | 4.196 | x | | |

| # | Journal | | | | | |
|---|---|---|---|---|---|---|
| 6. | Earth and Planetary Science Letters | 16 | 4.734 | x | | |
| 7. | Pure and Applied Geophysics | 15 | 1.618 | x | | |
| 8. | Nature | 14 | 41.456 | x | | x |
| 9. | Journal of Disaster Research | 14 | SJR 0.18 | | x | |
| 10. | Journal of Geophysical Research: Solid Earth | 12 | 3.426 | x | | |
| 11. | International Journal of Disaster Risk Reduction | 12 | SJR 0.510 | | x | x |
| 12. | Bulletin of the International Institute of Seismology and Earthquake Engineering | 12 | SJR 0.12 | x | | |

**Table 5: Comparing citations authored in general and those first authored by Indonesian in 10 most cited papers (source: modified from SCOPUS results)**

Note: Y=Year, J=Journal, C=Number of Citations, IF=Journal impact factors, I=Indonesia author (marked at the authors column)

| Overall | | | | | | First authored by Indonesian | | | | | |
|---|---|---|---|---|---|---|---|---|---|---|---|
| Authors (Indonesian are marked I) | Title | Y | J | C | IF | Authors (Indonesian are marked I) | Title | Y | J | C | IF |
| Page S.E., Siegert F., Rieley J.O., Boehm H.-D.V., Jaya A., (I) Limin S. (I) | The amount of carbon released from peat and forest fires in Indonesia during 1997 | 2002 | Nature | 1280 | 41.456 | Aldrian E. (I), Susanto RD (I) | Identification of three dominant rainfall regions within Indonesia and their relationship to sea surface temperature | 2003 | International Journal of Climatology | 344 | 3.609 |
| Siegert F., Ruecker G., Hinrichs A., Hoffmann A.A. | Increased damage from fires in logged forests during droughts caused by El Niño | 2001 | Nature | 519 | 41.456 | Subarya, C (I), Chlieh, M, Prawirodirdjo, L (I), Avouac, JP, Bock, Sieh, Meltzner, Natawidjaja (I), McCaffrey | Plate-boundary deformation associated with the great Sumatra-Andaman earthquake | 2006 | Nature | 343 | 41.456 |
| Ishii M., Shearer P.M., Houston H., Vidale J.E. | Extent, duration and speed of the 2004 Sumatra-Andaman earthquake imaged by the Hi-Net array | 2005 | Nature | 386 | 41.456 | Susanto RD. (I), Gordon A.L., Zheng Q. | Upwelling along the coasts of Java and Sumatra and its relation to ENSO | 2001 | Geophysical Research Letters | 161 | 4.196 |
| Aldrian E. (I), Dwi Susanto R. (I) | Identification of three dominant rainfall regions within Indonesia and their relationship to sea surface temperature | 2003 | International Journal of Climatology | 343 | 3.157 | Natawidjaja, DH (I), Sieh, K., Chlieh, M.,, Galetzka, J.,, Suwargadi, BW., (I), Cheng, H., Edwards, RL., Avouac, JP., Ward, SN | Source parameters of the great Sumatran megathrust earthquakes of 1797 and 1833 inferred from coral microatolls | 2006 | Journal of Geophysical Research: Solid Earth | 156 | 3.318 |
| Subarya, C (I), Chlieh, M, Prawirodirdjo, L (I), Avouac, JP, | Plate-boundary deformation associated with the great Sumatra- | 2006 | Nature | 343 | 41.456 | Danny Hilman Natawidjaja (I), Kerry Sieh, Steven N Ward, | Paleogeodetic records of seismic and aseismic subduction from central Sumatran | 2004 | Journal of Geophysical Research | 119 | 3.318 |

| Overall | | | | | | First authored by Indonesian | | | | | |
|---|---|---|---|---|---|---|---|---|---|---|---|
| Authors (Indonesian are marked I) | Title | Y | J | C | IF | Authors (Indonesian are marked I) | Title | Y | J | C | IF |
| Bock, Sieh, Meltzner, Natawidjaja (I), McCaffrey | Andaman earthquake | | | | | Hai Cheng, R Lawrence Edwards, John Galetzka, Bambang W Suwargadi (I) | microatolls, Indonesia | | : Solid Earth | | |
| Rampino MR., Self S. | Volcanic winter and accelerated glaciations following the Toba super-eruption | 1992 | Nature | 333 | 41.456 | Abidin HZ, Djaja, R, Darmawan D, Hadi. S, Akbar, S, Rajiyowiryono, Sudibyo, Y, Meilano, I, Kasuma, MA, Kahar, J, Subarya, C (All I) | Land subsidence of Jakarta (Indonesia) and its geodetic monitoring system | 2001 | Natural Hazards | 103 | 1.719 |
| Sieh, Natawidjaja (I) | Neotectonics of the Sumatran fault, Indonesia | 2000 | Journal of Geophysical Research: Solid Earth | 317 | 3.426 | Andreastuti S.D. (I), Alloway B.V., Smith I.E.M. | A detailed tephrostratigraphic framework at Merapi Volcano, Central Java, Indonesia: Implications for eruption predictions and hazard assessment | 2000 | Journal of Volcanology and Geothermal Research | 81 | 2.543 |
| C Vigny, WJF Simons, S Abu, R Bamphenyu, C Satirapod, N Choosakul, C Subarya, A Socquet, K Omar, HZ Abidin, BAC Ambrosius | Insight into the 2004 Sumatra–Andaman earthquake from GPS measurements in southeast Asia | 2005 | Nature | 329 | 41.456 | Marfai, M. A. (I), King, L | Monitoring land subsidence in Semarang, Indonesia | 2007 | Environmental Geology Journal of Geophysical Research : Solid Earth | 68 | 3.318 |
| Hsu Y.-J., Simons M., Avouac J.-P., Galeteka J., Sieh K., Chlieh M., Natawidjaja D. (I), Prawirodirdjo L. (I), Bock Y. | Frictional afterslip following the 2005 Nias-Simeulue earthquake, Sumatra | 2006 | Science | 271 | 33.61 | Marfai, M. A. (I), King, L | Potential vulnerability implications of coastal inundation due to sea level rise for the coastal zone of Semarang city, Indonesia | 2008 | Environmental Geology Journal of Geophysical Research : Solid Earth | 59 | 3.318 |
| Briggs R.W., Sieh K., Meltzner A.J., Natawidjaja D. (I), Galetzka J., Suwargadi B. (I), Hsu Y.-J., Simons M., Hananto N. (I), Suprihanto I. (I), Prayudi D. (I), Avouac J.-P., Prawirodirdjo L. (I), Bock Y. | Deformation and slip along the Sunda megathrust in the great 2005 Nias-Simeulue earthquake | 2006 | Science | 226 | 33.61 | Marfai MA (I),, Almohammad H, Sudip Dey, Susanto, B (I),, King, L | Coastal dynamic and shoreline mapping: multi-sources spatial data analysis in Semarang Indonesia | 2008 | Environmental Monitoring and Assessment | 57 | 1.663 |

| Overall | | | | | | First authored by Indonesian | | | | | |
|---|---|---|---|---|---|---|---|---|---|---|---|
| Authors (Indonesian are marked I) | Title | Y | J | C | IF | Authors (Indonesian are marked I) | Title | Y | J | C | IF |
| Konca A.O., Avouac J.-P., Sladen A., Meltzner A.J., Sieh K., Fang P., Li Z., Galetzka J., Genrich J., Chlieh M., Natawidjaja DH. (I), Bock Y., Fielding E.J., Ji C., Helmberger D.V. | Partial rupture of a locked patch of the Sumatra megathrust during the 2007 earthquake sequence | 2008 | Nature | 207 | 41.456 | Amien I. (I), Rejekiningrum P. (I), Pramudia A. (I), Susanti E (I). | Effects of interannual climate variability and climate change on rice yield in Java, Indonesia | 1996 | Water, Air, and Soil Pollution | 51 | 1.554 |
| **Total** | | | | **4547** | **296.775** | | | | | **1542** | **70,012** |

## List of Figures

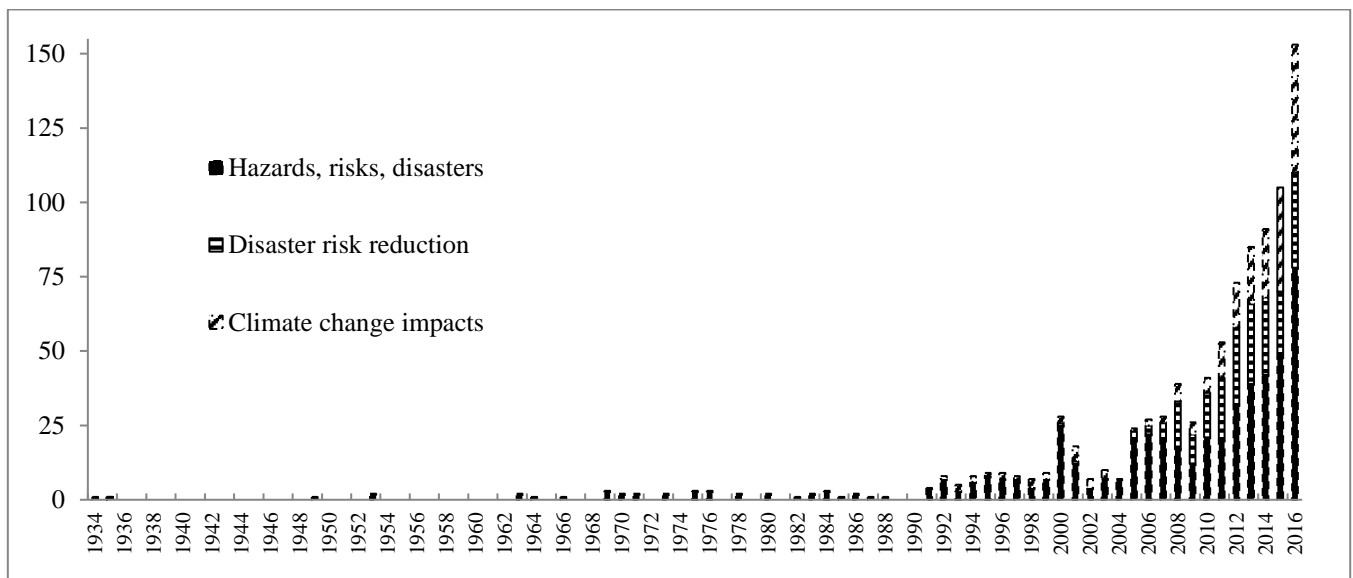

**Figure 1: Number of publications over the year (modified from SCOPUS, 2016a)**

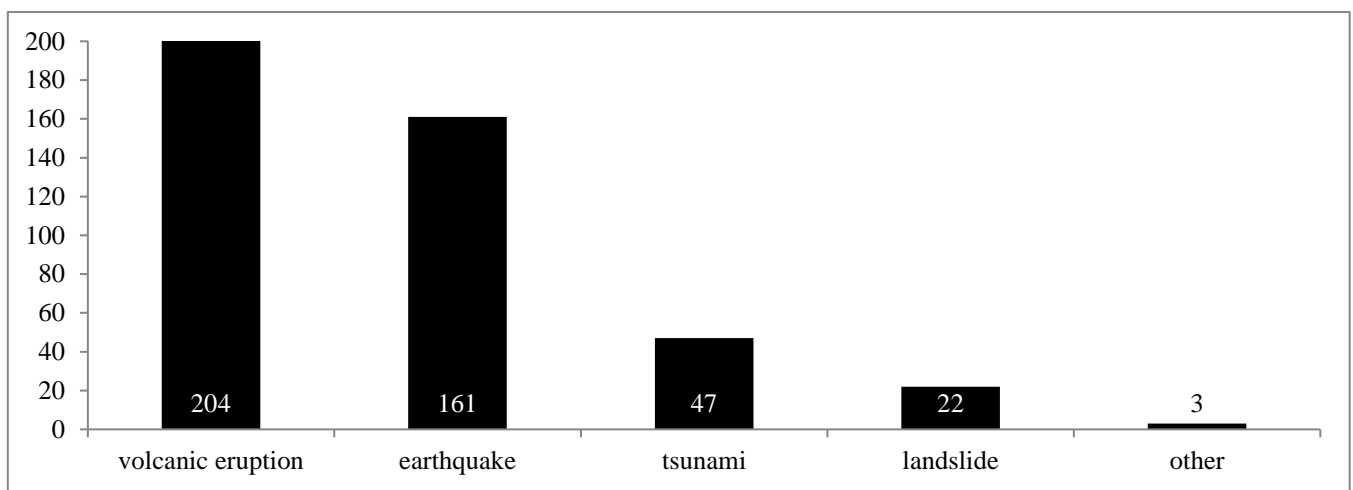

**Figure 2: Key topics in HRD category (Source; modified from SCOPUS results)**

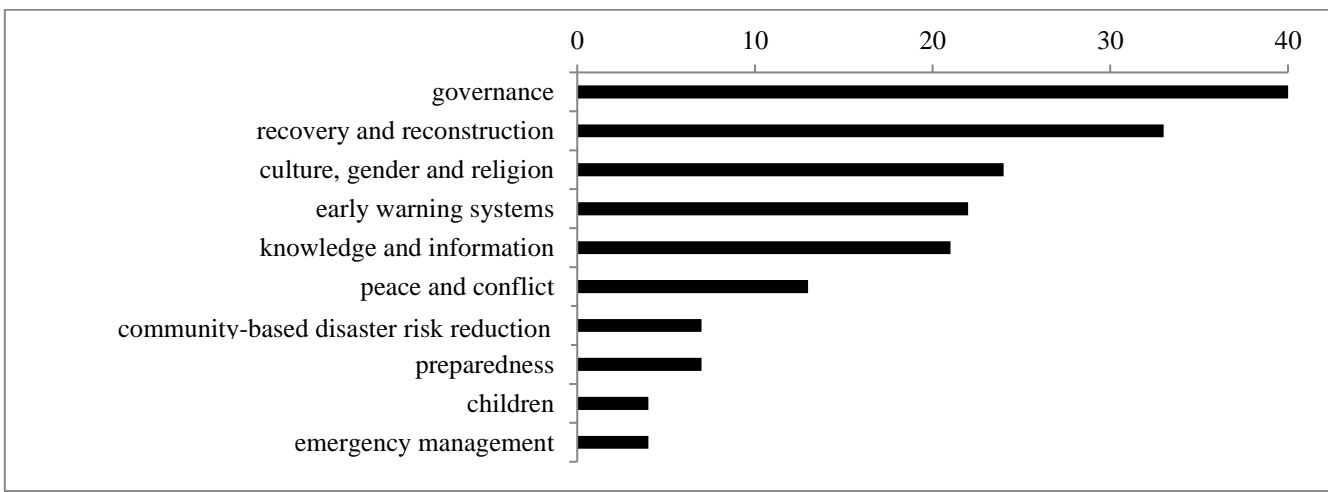

**Figure 8: Key topics in DRR category (Source; modified from SCOPUS results)**

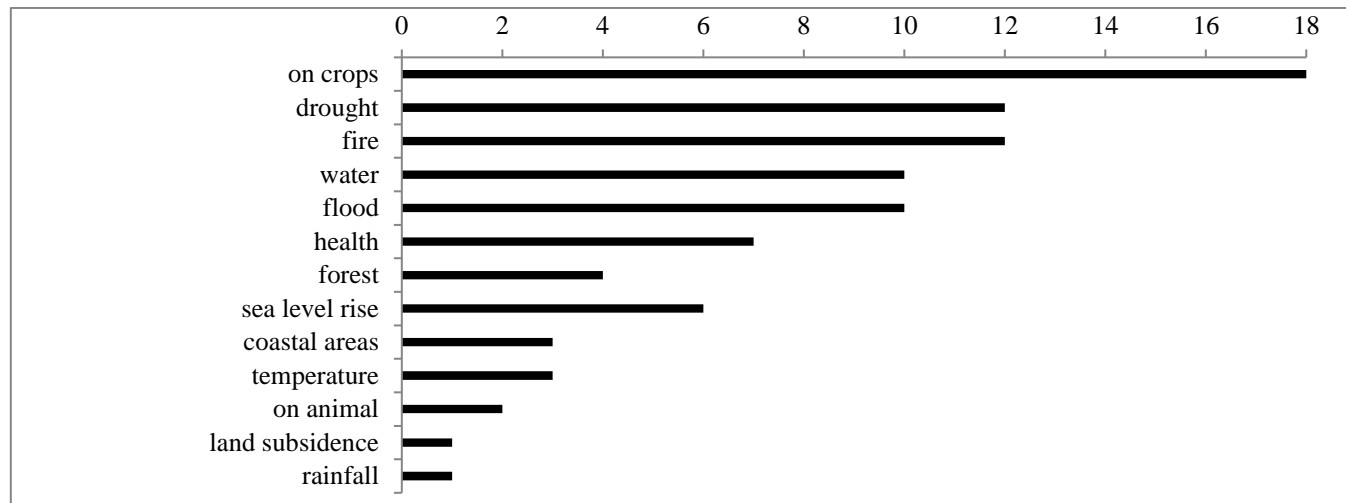

**Figure 3: Key topics in CC category researching on impacts of climate change (Source: modified from SCOPUS results)**

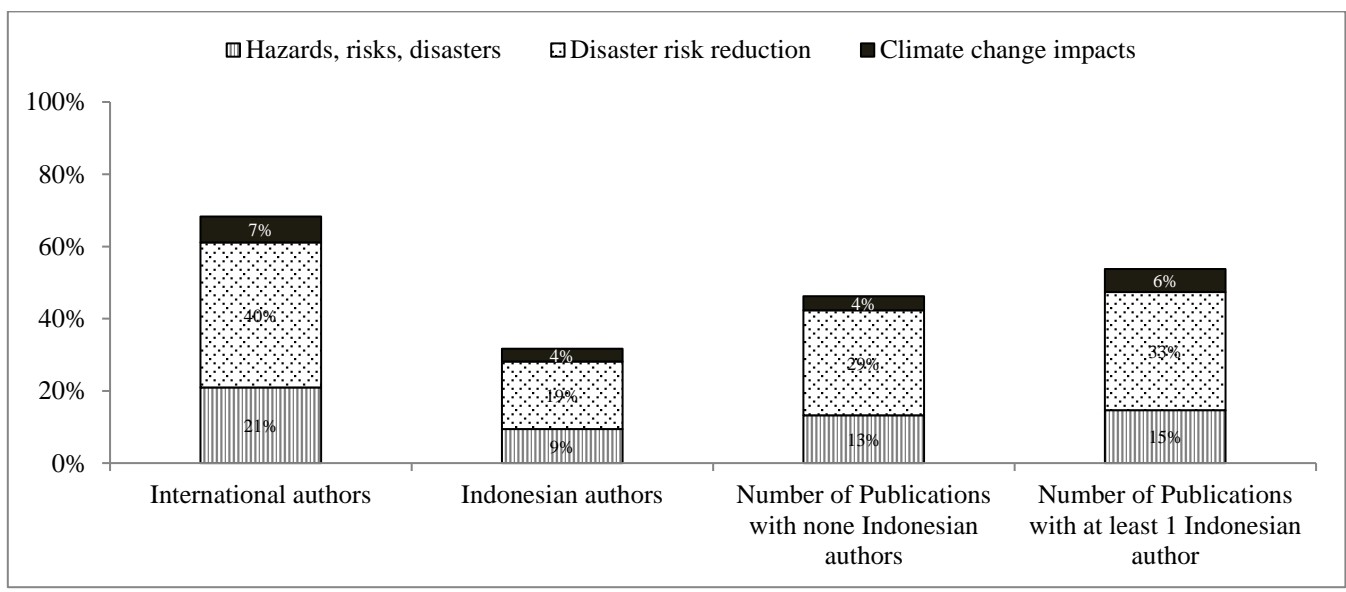

**Figure 9: Comparing the roles of international and Indonesian authors in each publication category (source: modified from SCOPUS results)**

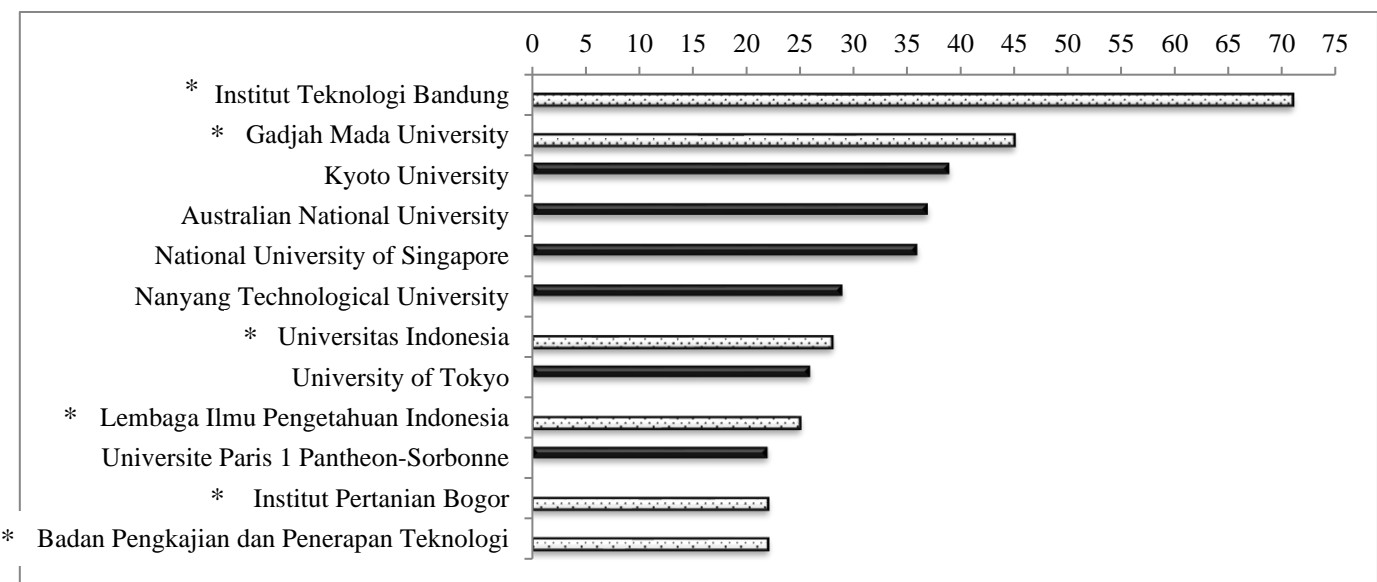

**Figure 5: Organizations with highest number of publications (Indonesian Organizations marked with *) (source: modified from SCOPUS results)**

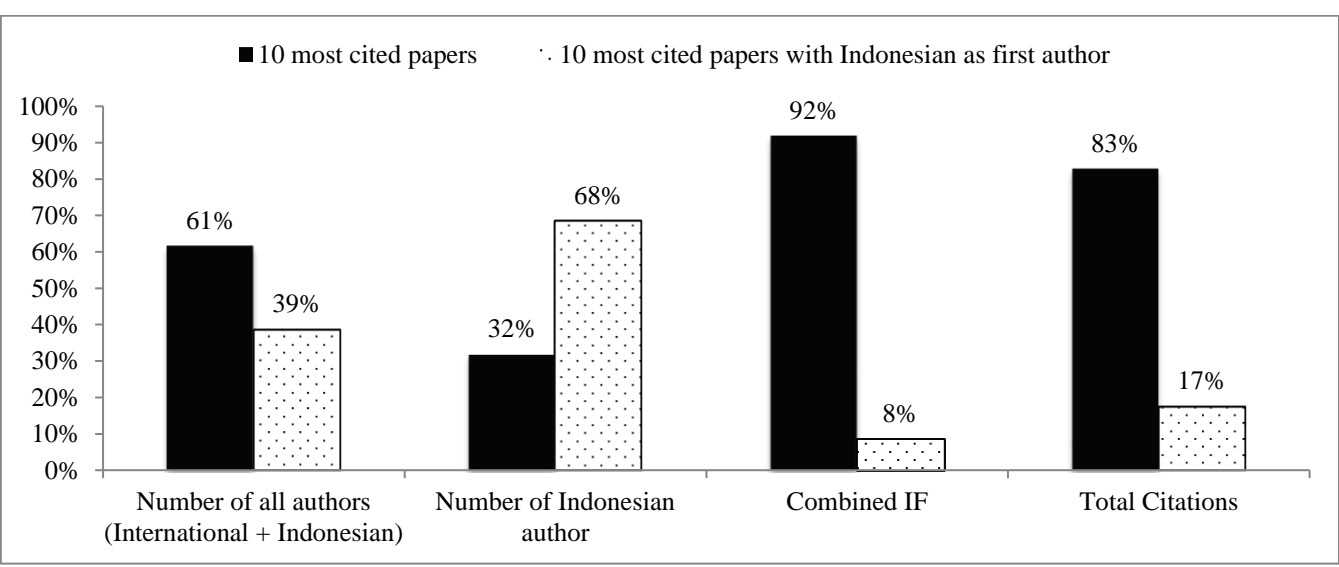

