# Peer review of "Review Article: a systematic literature review of research trends and authorships on natural hazards, disasters, risk reduction and climate change in Indonesia"

_Natural Hazards and Earth System Sciences, 2016_

## Referee Comment (RC1) · Anonymous Referee #1 · 22 Nov 2016

The paper presents a systematic literature review on the research trends on Natural Hazards, Disaster Risk Reduction and Climate Change in Indonesia. I think that this paper has some potential but need some major reworking before publication. I hope that all the suggestions and comment raised will improve the quality of the manuscript.

Abstract: The timeline considered isn't it from 1977 as expressed later in the manuscript (line 147; pg 6)?

Introduction: The author reviews three main topics (1) HDR (2) DRR and (3) CC, but the introduction just focuses on the presentation of disaster events in Indonesia and on the comparison between geophysical and hydro-meteor-climato-logical disasters (is this last word correct by the way?). This small introduction (lines 24-35) does not give a

clear picture of why the author undertook this review. The real understanding is given from lines 60-onwards where the author underlines the need of more data on DRR and the management of climate related hazards. In addition, I suggest to report the information of figure 2 directly in the text and therefore to remove figure 2. Furthermore, after recognizing the gaps of knowledge, the author can present the objectives of the paper. Accordingly, I suggest to finish the introduction section with "this papers aims to systematically review.." that is presented from line 38 onwards in pg.2.

Research method: The methodology has been undertaken correctly since few researchers explore the systematic literature review, as it is complex and time consuming. Anyway, the sub-chapter 2.1, 2.2, 2.2.1, 2.2.2 and 2.2.3 should be merged together into one main body under "2. Research methods". The chapter 2.3 "Analysis and presentation of results" seems a repetition of the small subchapter presented before for which I suggest merging this in the previous chapter.

Findings and Analysis: The categorization of disaster groups in table 4 is taken from EM-DAT 2016 and any added value is given. I suggest removing the table and integrating the citation in the text. When the author presents each topic (3.1.1-onwards), there is no need to explore them with so many small subchapters (timeline, discussions and focus areas). I suggest to merge these paragraphs trying to give an overall sequence and shape. The same for the topics 3.3.2 and 3.3.3. I think that there are some errors in the numbering of the chapters. Please revise it carefully. The discussion part presented from line 285 to 297 pg. 10-11 is not discussed at all. Please provide some ideas and key points on it. In its form it is a mere list. The "Progress of Indonesian researchers and organization" chapter is valuable and I personally think that is the core of the paper, never explored in literature before. However, I think that this chapter could be presented without so many subchapters and the author needs to rearrange it merging the information into one main section. Please, try to present the results of this chapter without coping and pasting the paragraphs in chronologically way they are presented now. In addition, the text in lines 328-334 seems an advertisement of first

[Figure]

authors as expressed in table 5. I would delete this information or rearranged the way it is presented.

Table 5. Please report in the caption all the acronyms used (e.g PVMBG, ITB etc). I cannot understand the symbol put after SC, GS, RG. In addition, I would delete the column field "other profile" since does not give any added value. Does the author Surono have a name?

Table 8. I suggest deleting the division "all authors" and "Indonesians authors" and adding just a field "authors" with a parenthesis indicating if from Indonesia (I) after the name. For the right table relating to the first authored by Indonesia I suggest doing something similar. Then, I would put IF of the journal after the name of the journal. Are columns "Journal" and "Journal name" the same? I suggest revising it carefully.

---

## Author Comment (AC1) · 19 Dec 2016

Dear Reviewer, The author would like to say thank you for the valuable comments given on this paper.

Kindly please find detailed responses to each issue raised in the review:

1. Comment: Abstract: The timeline considered isn't it from 1977 as expressed later in the manuscript (line 147; pg 6)?

Response: The sentence is revised Line 10: To address this, the author conducted a systematic literature review of related publications indexed within the SCOPUS database from 1900 to 2016.

2. Comment: The author reviews three main topics (1) HDR (2) DRR and (3) CC, but the introduction just focuses on the presentation of disaster events in Indonesia and on the comparison between geophysical and hydro-meteor-climato-logical disasters (is this last word correct by the way?). This small introduction (lines 24-35) does not give a clear picture of why the author undertook this review.

Response: Line 32: The aim is rewritten This paper aims to systematically review literature related to natural hazards, risk and disaster risk reduction, along with the strategies to mitigate and manage the events and impacts, as well as climate change vulnerability, impact, and assessments in Indonesia.

3. Comment: I suggest to report the information of figure 2 directly in the text and therefore to remove figure 2.

Response: Figure 2 is removed

4. Comment: Furthermore, after recognizing the gaps of knowledge, the author can present the objectives of the paper. Accordingly, I suggest to finish the introduction section with "this papers aims to systematically review.." that is presented from line 38 onwards in pg.2. Response: Line 30-55 show the elaborations on the aim of the paper.

5. Comment: Research method: The methodology has been undertaken correctly since few researchers explore the systematic literature review, as it is complex and time consuming. Anyway, the sub-chapter 2.1, 2.2, 2.2.1, 2.2.2 and 2.2.3 should be merged together into one main body under "2. Research methods". The chapter 2.3 "Analysis and presentation of results" seems a repetition of the small subchapter presented before for which I suggest merging this in the previous chapter.

Response: The author reorganize this section completely, and to have only sub-headings 2.1 on Data Collection, and and 2.2 on Data Analysis.

6. Comment: Findings and Analysis: The categorization of disaster groups in table 4 is taken from EM-DAT 2016 and any added value is given. I suggest removing the table

and integrating the citation in the text. When the author presents each topic (3.1.1-onwards), there is no need to explore them with so many small subchapters (timeline, discussions and focus areas). I suggest to merge these paragraphs trying to give an overall sequence and shape. The same for the topics 3.3.2 and 3.3.3. I think that there are some errors in the numbering of the chapters. Please revise it carefully. The discussion part presented from line 285 to 297 pg. 10-11 is not discussed at all. Please provide some ideas and key points on it. In its form it is a mere list.

Response: The author also simplified this section, and the small subchapters are merged.

6. Comment: The "Progress of Indonesian researchers and organization" chapter is valuable and I personally think that is the core of the paper, never explored in literature before. However, I think that this chapter could be presented without so many sub-chapters and the author needs to rearrange it merging the information into one main section. Please, try to present the results of this chapter without coping and pasting the paragraphs in chronologically way they are presented now. In addition, the text in lines 328-334 seems an advertisement of first authors as expressed in table 5. I would delete this information or rearranged the way it is presented.

Response: Section 3.2 is also simplified.

6. Table 5. Please report in the caption all the acronyms used (e.g PVMBG, ITB etc). I cannot understand the symbol put after SC, GS, RG. In addition, I would delete the column field "other profile" since does not give any added value. Does the author Surono have a name?

Response: All acronims have been spelt out when they first used. Column on other profile is deleted Surono is a one name author as stated in Scopus

Please also note the supplement to this comment:
http://www.nat-hazards-earth-syst-sci-discuss.net/nhess-2016-342/nhess-2016-342-

AC1-supplement.pdf

**Supplement:**

**Research Trends in Natural Hazards, Disasters, Risk Reduction and Climate Change in Indonesia: A Systematic Literature Review**

Riyanti Djalante[1, 2]

[1]United Nations University – Institute of Environment and human Security, Bonn, 53117, Germany
5   [2]University of Halu Oleo, Kendari, 93111, Sulawesi Tenggara, Indonesia

*Correspondence to*: Djalante@ehs.unu.edu

**Abstract.** Indonesia is one of the most vulnerable countries from disasters and climate change. While there has been a proliferation of academic publications written on issues related to natural hazards, risks, and disasters on Indonesia, there has not yet a systematic literature review (SLR) to determine the progress, key topics and directions for further research. SLR is
10   important so researchers can build upon existing works, avoid bias, determine major research and need for further research. It is also important to determine who, how, in which way the research has been conducted in order to strengthen research capacity in the future. The author conducted a SLR of publications indexed within the Scopus database from 1900 to 2016 on topics related to natural hazards, risks, risk reduction and climate change impacts on Indonesia. The findings are outlined in two parts. The first part focuses on the research topics and finds that publications can be categorized into three major
15   topics: (1) natural hazard, risk and disaster assessments (HRD), (2) disaster risk reduction (DRR), and (3) climate change risks, vulnerability, impacts and adaptation (CC). More than half the publications fall into HRD and focus on volcanic eruptions, tsunami and earthquakes. Publications on DRR focus on governance, early warning systems and recovery and reconstruction. Those regarding CC mainly concern carbon emission, forestry, governance, and impacts. The second part focuses on roles of Indonesian researchers and organizations in these publications. Findings show limited progress in
20   research, publication and collaboration. International/ non-Indonesian authors dominate the literature and only half of the publications are co-authored by Indonesians. Moreover, of the international collaborations that took place, this was limited to only a few Indonesian organizations. Reasons for this could be limited experience in academic collaboration, power play amongst researchers, lack of research capacity, weak English academic writings skills as well as a lack of incentives for international collaboration and publication within the Indonesian higher education system.

**Keyword: Systematic literature review; natural hazard; disaster; climate change; Indonesia**

**1 Introduction**

Disaster events and their associated social and economical impacts are on the rise (EMDAT, 2016). The last decade has witnessed the highest number and impacts from disasters and 2015 has been declared as the hottest year ever (WMO, 2016). The Asia Pacific region has experienced the highest number of disasters on record (EMDAT, 2016), within which Indonesia is one of the most at risk countries to disasters and climate change impacts (EMDAT, 2016). Between the period of 1900 to 2016, there have been a total of 434 disasters in Indonesia caused by natural hazards, with 237,728 deaths, 29.1 million people affected and total damage almost 30 Billion USD (EMDAT, 2016). Geophysical hazards caused more than 90% deaths while the hydrometerological occur more frequently, affected more people, and caused three times damages (EMDAT, 2016). This paper aims to systematically review literature related to natural hazards, risk and disaster risk reduction, as well as climate change vulnerability, impact, and assessments in Indonesia. A systematic literature review (SLR) is defined as a method for systematically reviewing evidence or literature with explicit and transparent methods (Gill and Malamud, 2014). Even though there is vast material on these topics on Indonesia, there has not yet been a literature review that examines them in a comprehensive and systematic way. By reviewing published works in this fashion, researchers can build upon others´ works, avoid bias (Khan et al., 1996) and reinventing the wheel so that topics that have been heavily researched can be determined, and those that need further research can be outlined (Moher et al., 2009b). It is also important to gauge who, how and in which way the research has been conducted, and determining this will enable consideration for strengthening research capacity in the future (Mallett et al., 2012).

There are two research aims adopted. The **first** is to determine progress of research in natural hazards, risks, disasters and climate change in Indonesia within the timeframe from 1900 to 2016. The **second** is to examine roles of Indonesian authors in contributing to research, international publications and collaborations. The importance of conducting literature on these topics is manifold. The Sendai Framework for DRR (SFDRR) has just been adopted and with it an extension of the scope of hazards and risk reduction strategies (UN/ISDR, 2015). There is a move toward an integrated approach to DRR which calls for strategies and actions to reduce risks and associated impacts, as well as an inclusive role of multiple actors in DRR. This review will enable the identification of strategies that have been undertaken for DRR and hence suggest strategies for future DRR and implementing the SFDRR. Also, there is an increasing focus on the impacts of climate change in the changing profile of hazards and disasters, and hence this calls for integrated DRR and climate change adaptation (CCA) to manage climate risks. This review will try to capture whether consideration of climate change risks have been considered as part of research progress in Indonesia. This study attempts to determine whether progress towards more specific studies on the national and local level is observable. Moreover, determining the progress of Indonesian scholars is important and relevant for several reasons. These 
[revised manuscript text omitted]

**2.2 Data Analysis**

The author used Scopus features to analyze search results such as the article metric module, citation overview, and author profile pages (SCOPUS, 2016b). This final list was analyzed in terms of authorship, references, citations, keywords, places of focus, types and time of publications, impact factors and topics and sub-topics of research. The progress of Indonesian scholars is evaluated through counting total number of authors, research outputs and citations overall, and also comparing between papers first authored by Indonesians. The author cross-checked the number of citations from Scopus on the Internet through Google, and selected the higher citation counts. This was done because it is generally the case that data from a Google search for a publication and author leads to a higher and more up to date citation count. The author also consulted total citations and publications of researchers in Google Scholar, Research Gate or from other websites to make sure that the full list of publications was captured. There were also cases where the author had to specifically go back to Scopus and find particular author's works to make sure that all were captured.

**3 Findings and Analysis**

This section is structured into two main parts, first with research topics, and second with progress of Indonesian researchers and organizations.

**3.1 Timelines and Research Topics**

This part presents the more detailed findings of each of the research topics. The author categorizes the final list into three groups (Table 2), natural hazard, risk, disaster assessments (HRD), disaster risk management and reduction (DRR), and climate change vulnerability, impacts and adaptation (CC), in order to show and outline how changes in directions on research have taken place over the years and to reduce unbalance towards findings on hazard and risks assessments toward earthquake and volcanic eruption research. There are 56% of HRD, and the rest is shared almost equally by the DRR and CC literature (modified from SCOPUS, 2016a).

**Table 2 Classifications of Findings Based on Topics of Research**

| Major topics groups | Definitions (IPCC, 2012; UNISDR, 2009) | Number of publications |
|---|---|---|
| (1) Natural hazard, risks, disasters assessments (HRD) | Hazards: A dangerous phenomenon, substance, human activity or condition that may cause loss of life, injury or other health impacts, property damage, loss of livelihoods and services, social and economic disruption, or environmental damage.
Risks: The combination of the probability of an event and its negative consequences.
Disaster: A serious disruption of the functioning of a community or a society involving widespread human, material, economic or environmental losses and impacts, which exceeds the ability of the affected community or society to cope using its own resources. | 517 |
| (2) disaster risk management or reduction (DRR) | The systematic process of using administrative directives, organizations, and operational skills and capacities to implement strategies, policies and improved coping capacities in order to lessen the adverse impacts of hazards and the possibility of disaster (UNISDR).
The concept and practice of reducing disaster risks through systematic efforts to analyze and manage the causal factors of disasters, including through reduced exposure to hazards, lessened vulnerability of people and property, wise management of land and the environment, and improved preparedness for adverse events. | 210 |
| (3) climate change vulnerability, impacts and adaptation (CC) | A change of climate which is attributed directly or indirectly to human activity that alters the composition of the global atmosphere and which is in addition to natural climate variability observed over comparable time periods (UNFCCC).
The adjustment in natural or human systems in response to actual or expected climatic stimuli or their effects, which moderates harm or exploits beneficial opportunities (UNISDR). | 194 |
| **Total** | | 921 |

The paper identifies key periods and timelines by which publications were published. In general, there are more research on the topic of HRD, followed by those in DRR, and then CC. The publications on the HRD are also some of the earliest publications indexed in Scopus. Although the search timeline was set between 1900 and 2016, the years in which publications were found ranges from 1934 to 2016 (Figure 1).

[Figure]

**Figure 1 Number of Publications over the Year (modified from SCOPUS, 2016a)**

145    The first period is from the 1934-1990s. There were no significant changes in the numbers of publications produced. Research in this period was heavily focused on the topics of geophysical hazards and risks related to earthquakes and volcanic eruptions  (SCOPUS, 2016a). Within this period, 22 out of 58 events recorded by EMDAT were earthquakes and volcanic activities (EMDAT, 2016). The Bali earthquakes occurred in 1976 and 1979, which in total caused 1764 deaths, affected 563,150 people, and caused 215,150 USD in damages (EMDAT, 2016). The year 1979 was also the year in which

150    the earthquake occurred the most (6 times), in Bali, Lombok, and Biak (near Papua) (USGS, 2016). The second period from the 1990s to 2000s shows a notable increase in the literature, up to an average there 10 publications per year. This gradual increase mainly corresponds to a rise in literature related to the assessments of hazards, risks and disasters, and is followed by a sharp increase in literature to its highest point in 2000 (SCOPUS, 2016a). The third period from 2000s-2010s was the most dynamic period for publications. While there was a sharp decline since it first peak in 2000, a surge of publications

155    begun in 2004 in response to the Indian Ocean tsunami which devastated Indonesia especially. This increase has continued ever since. This is also a period characterized not only publications related to understanding the risks of earthquakes and tsunami, but also those related to DRR and CC. A peak occurs between 2010 and 2016 which shows soaring published materials in all topics. There were 153 publications in 2016 which is the highest ever produced in a single year. During this period, publications related to climate change and its impact on Indonesia has started to be considered and is expected to rise

160    further in the future. Both publications on HRD and CC are expected to rise (SCOPUS, 2016a).

The following sub-sections outline research issues discussed within the three topic groups. Within each, the paper discusses timelines, focus areas of the research, early contributors, and categorization of key topics discussed.

**3.1.1 Natural hazards, risks and disasters assessments (HRD)**

The first sub-section explains findings on the topic of hazards, risks and disasters assessments and identifications. The EMDAT-CRED (2016) categorization of HRD that is used in this study to help more detailed analysis related to major research topics. Natural-disaster groups caused by geophysical, meteorological, hydrological, and climatologically hazards are included since it is determined that these are the most frequent and impactful disasters in the country. Those excluded are disasters caused by biological, extraterrestrial and technological hazards.

There are 535 publications in this category (SCOPUS, 2016a). The findings show that there has been a gradual increase in the number of published materials from 1934 to 2000. It first reached its first peak in 2000 that the research in this topic reached its first significant outputs of 25 publications, and reduced slightly after that. In 2004 the Indian Ocean tsunami occurred, initiated with the 9.8 M earthquake with the epicenter off the island of Sumatra, badly affecting Indonesian. Publications related to the tsunami continued to be published until it reached a peak in 2006. Then in 2009, the publications have increased rapidly ever since, reaching another peak in 2015 of 47 publications in a single year (SCOPUS, 2016a). The islands of Java and Sumatera are the two areas which receive most attention (more than 70%) (SCOPUS, 2016a). The studies in these two islands are mostly related to the study of volcanic eruptions, earthquakes and tsunami. This is not surprising considering that Indonesia has the most numbers of volcanoes and is located along the Pacific ring of fire where earthquakes occur the most (USGS, 2016). The island of Sumatera directly experienced and was impacted by one of the most powerful earthquakes of 8.9 R.S which caused the tsunami in 2004 and hit Aceh, in the north west of Sumatera (Ishii et al., 2005).

Most of the literature around this period focuses on the impacts of volcanic eruptions in Java and Sumatera. The oldest publications related to HRD in Indonesia 
[revised manuscript text omitted]
. Studies on the roles of international and local authorships and collaborations show that although it rapidly increasing, there are still more efforts needed to strengthen and advance those collaborations (Bordons et al., 1996; Wagner and Leydesdorff, 2005b, a; Gazni et al., 2012). It further shows that there is still imbalance in the ratio of male to female scientists, as the global trends also show (Sidhu et al., 2009; Lewison, 2001; Koppel et al., 2002; Sugimoto et al., 2013). The importance of science communication and the increasing demand for researchers to publish their works outside of traditional methods such as journal articles, but also through blogs, websites, policy briefs, and popular media is now encouraged (Gu and Widén-Wulff, 2011; Thelwall et al., 2013; Bik and Goldstein, 2013).

**3.2.1 Authorships**

[revised manuscript text omitted]

Keil, A., Zeller, M., Wida, A., Sanim, B., and Birner, R.: What determines farmers' resilience towards ENSO-related drought? An empirical assessment in Central Sulawesi, Indonesia, Climatic Change, 86, 291-307, 10.1007/s10584-007-9326-4, 2008.

765    Keil, A., Teufel, N., Gunawan, D., and Leemhuis, C.: Vulnerability of smallholder farmers to ENSO-related drought in Indonesia, Climate Research, 38, 155-169, 10.3354/cr00778, 2009.

Kelman, I.: Tsunami diplomacy: Will the 26 December, 2004 bring peace to the affected countries?, Sociological Research Online, 10, 2005.

Khan, K. S., Daya, S., and Jadad, A. R.: THe importance of quality of primary studies in producing unbiased systematic 770    reviews, Archives of Internal Medicine, 156, 661-666, 1996.

Khandekar, M. L., Murty, T. S., Scott, D., and Baird, W.: The 1997 El Nino, Indonesian Forest fires and the Malaysian Smoke problem: A deadly combination of natural and man-made hazard, Natural Hazards, 21, 131-144, 2000.

Kitchenham, B., Pearl Brereton, O., Budgen, D., Turner, M., Bailey, J., and Linkman, S.: Systematic literature reviews in software engineering - A systematic literature review, Information and Software Technology, 51, 7-15, 10.1016/j.infsof.2008.09.009, 2009.

775 Kõlves, K., Kõlves, K. E., and De Leo, D.: Natural disasters and suicidal behaviours: A systematic literature review, Journal of Affective Disorders, 146, 1-14, 10.1016/j.jad.2012.07.037,

Konca, A. O., Avouac, J. P., Sladen, A., Meltzner, A. J., Sieh, K., Fang, P., Li, Z., Galetzka, J., Genrich, J., Chlieh, M., Natawidjaja, D. H., Bock, Y., Fielding, E. J., Ji, C., and Helmberger, D. V.: Partial rupture of a locked patch of the Sumatra

780 megathrust during the 2007 earthquake sequence, Nature, 456, 631-635, 10.1038/nature07572, 2008.

Koppel, M., Argamon, S., and Shimoni, A. R.: Automatically Categorizing Written Texts by Author Gender, Literary and Linguistic Computing, 17, 401-412, 10.1093/llc/17.4.401, 2002.

Koshimura, S., Oie, T., Yanagisawa, H., and Imamura, F.: Developing fragility functions for tsunami damage estimation using numerical model and post-tsunami data from banda aceh, Indonesia, Coastal Engineering Journal, 51, 243-273,

785 10.1142/s0578563409002004, 2009.

Kram, K. E., and Isabella, L. A.: Mentoring alternatives: The role of peer relationships in career development, Academy of management Journal, 28, 110-132, 1985.

Kusumadinata, K.: Letusan Gunung Agung di Bali tahun 1963 (The eruption of the Agung volcano in Bali, in 1963), Geological Survey of Indonesia, Bandung, 1963.

790 Kusumadinata, K.: Letusan Gunung Agung di Bali tahun 1963 (The eruption of the Agung volcano in Bali, in 1963), Bulletin of Geological Survey Indonesia, 1, 12-15, 1964a.

Kusumadinata, K.: Lanjutan Kegiatan Gunung Agung bulan Januari 1965 (Renewed activity of the Agung volcano in January 1964), Bulletin of Geological Survey Indonesia, 1, 38, 1964b.

Kusumadinata, K.: Lahars of the Agung volcano as a secondary destructive element, Bulletin of Geological Survey

795 Indonesia, 1, 1964c.

Kusumasari, B., Alam, Q., and Siddiqui, K.: Resource capability for local government in managing disaster, Disaster Prevention and Management: An International Journal, 19, 438-451, 2010.

Kusumasari, B., and Alam, Q.: Bridging the gaps: The role of local government capability and the management of a natural disaster in Bantul, Indonesia, Natural Hazards, 60, 761-779, 10.1007/s11069-011-0016-1, 2012.

800 Larivière, V., Ni, C., Gingras, Y., Cronin, B., and Sugimoto, C. R.: Bibliometrics: Global gender disparities in science, Nature, 504, 2013.

Larson, S., Alexander, K. S., Djalante, R., and Kirono, D. G. C.: The Added Value of Understanding Informal Social Networks in an Adaptive Capacity Assessment: Explorations of an Urban Water Management System in Indonesia, Water Resources Management, 27, 4425-4441, 10.1007/s11269-013-0412-2, 2013.

805 Lassa, J. A.: Disaster Policy Change in Indonesia 1930-2010: From Government to Governance?, International Journal of Mass Emergencies & Disasters, 31, 2013.

Lassa, J. A.: Post disaster governance, complexity and network theory, PLoS Currents, 7, 10.1371/4f7972ecec1b6, 2015.

Lassa, J. A., and Nugraha, E.: From shared learning to shared action in building resilience in the city of Bandar Lampung, Indonesia, Environment and Urbanization, 27, 161-180, 10.1177/0956247814552233, 2015.

810 Latter, J. H.: Tsunamis of volcanic origin: Summary of causes, with particular reference to Krakatoa, 1883, Bulletin Volcanologique, 44, 467-490, 10.1007/bf02600578, 1981.

[revised manuscript text omitted]

Natawidjaja, D. H., Sieh, K., Ward, S. N., Cheng, H., Edwards, R. L., Galetzka, J., and Suwargadi, B. W.: Paleogeodetic

880 records of seismic and aseismic subduction from central Sumatran microatolls, Indonesia, Journal of Geophysical Research: Solid Earth, 109, 2004.

Natawidjaja, D. H., Sieh, K., Chlieh, M., Galetzka, J., Suwargadi, B. W., Cheng, H., Edwards, R. L., Avouac, J. P., and Ward, S. N.: Source parameters of the great Sumatran megathrust earthquakes of 1797 and 1833 inferred from coral microatolls, Journal of Geophysical Research: Solid Earth, 111, 2006.

885 Naylor, R. L., Falcon, W. P., Rochberg, D., and Wada, N.: Using El Niño/Southern Oscillation climate data to predict rice production in Indonesia, Climatic Change, 50, 255-265, 10.1023/a:1010662115348, 2001.

Neale, T., and Weir, J. K.: Navigating scientific uncertainty in wildfire and flood risk mitigation: A qualitative review, International Journal of Disaster Risk Reduction, 13, 255-265, http://dx.doi.org/10.1016/j.ijdrr.2015.06.010, 2015.

Neolaka, A.: Flood disaster risk in Jakarta, Indonesia, WIT Transactions on Ecology and the Environment, 159, 107-118,

890 10.2495/friar120091, 2012.

Neolaka, A.: Stakeholder participation in flood control of Ciliwung river, Jakarta, Indonesia, WIT Transactions on Ecology and the Environment, 171, 275-285, 10.2495/wrm130251, 2013.

Nicholls, R. J., Mimura, N., and Topping, J. C.: Climate change in south and south-east Asia: some implications for coastal areas, Journal of Global Environment Engineering, 1, 137-154, 1995.

895 OECD, and ADB: Reviews of National Policies for Education in Indonesia: Rising to the Challenge, 2015.

Page, S. E., Siegert, F., Rieley, J. O., Boehm, H. D. V., Jaya, A., and Limin, S.: The amount of carbon released from peat and forest fires in Indonesia during 1997, Nature, 420, 61-65, 10.1038/nature01131, 2002.

Philibosian, B., Sieh, K., Natawidjaja, D. H., Chiang, H. W., Shen, C. C., Suwargadi, B. W., Hill, E. M., and Edwards, R. L.: An ancient shallow slip event on the Mentawai segment of the Sunda megathrust, Sumatra, Journal of Geophysical

900 Research: Solid Earth, 117, 2012.

Prayoedhie, S., Fujii, Y., and Shibazaki, B.: Numerical simulations for Tsunami forecasting at Padang city using offshore Tsunami sensors, Bulletin of the International Institute of Seismology and Earthquake Engineering, 46, 97-102, 2012.

Purnomo, H., Herawati, H., and Santoso, H.: Indicators for assessing Indonesia's Javan rhino National Park vulnerability to climate change, Mitigation and Adaptation Strategies for Global Change, 16, 733-747, 10.1007/s11027-011-9291-0, 2011.

905 Volcanology Survey Indonesia: http://www.vsi.esdm.go.id/, access: March 4, 2016.

QS World University Rankings® 2015/16: http://www.topuniversities.com/university-rankings/world-university-rankings/2015#sorting=rank+region=+country=+faculty=+stars=false+search=, access: June 28, 2016.

Rafliana, I.: Disaster education in Indonesia: Learning how itworks from six years of experience after Indian ocean tsunami in 2004, Journal of Disaster Research, 7, 83-91, 2012.

910 Raleigh, C., Jordan, L., and Salehyan, I.: Assessing the impact of climate change on migration and conflict, Paper commissioned by the World Bank Group for the Social Dimensions of Climate Change workshop, Washington, DC, 2008, 5-6,

Rampino, M. R., and Self, S.: Historic eruptions of Tambora (1815), Krakatau (1883), and Agung (1963), their stratospheric aerosols, and climatic impact, Quaternary Research, 18, 127-143, http://dx.doi.org/10.1016/0033-5894(82)90065-5, 1982.

915 Rampino, M. R., and Self, S.: Volcanic winter and accelerated glaciation following the Toba super-eruption, Nature, 359, 50-52, 1992.

Early Career Researchers: http://www.rcuk.ac.uk/international/funding/fundingopps/earlycareer/, access: June 28, 2016.

Research Gate: https://www.researchgate.net/, access: March 4, 2016a.

Research Gate: https://www.researchgate.net/home, access: November 11, 2016b.

920 Reuveny, R.: Climate change-induced migration and violent conflict, Political Geography, 26, 656-673, 2007.

Riesenberg, D., and Lundberg, G. D.: The order of authorship: who's on first?, JAMA, 264, 1857-1857, 1990.

Sistem Informasi Penelitian dan Pengabdian Kepada Masyarakat: http://simlitabmas.ristekdikti.go.id/, access: June 28, 2016.

Rittmann, A.: Magmatic character and tectonic position of the Indonesia Volcanoes, Bulletin Volcanologique, 14, 45-58, 10.1007/bf02596004, 1953.

925 Sagala, S., Okada, N., and Paton, D.: Predictors of intention to prepare for volcanic risks in Mt Merapi, Indonesia, Journal of Pacific Rim Psychology, 3, 47-54, 2009.

Salafsky, N.: Drought in the rain forest: Effects of the 1991 El Niño-Southern Oscillation event on a rural economy in West Kalimantan, Indonesia, Climatic Change, 27, 373-396, 10.1007/bf01096268, 1994.

Sano, D., Prabhakar, S. V. R. K., Kartikasari, K., and Irawan, D. J.: Developing Adaptation Policies in the Agriculture

930 Sector: Indonesia's Experience, in: Climate Change Adaptation in Practice: From strategy development to implementation, 269-281, 2013.

Santosa, H.: Environmental management in Surabaya with reference to National Agenda 21 and the social safety net programme, Environment and Urbanization, 12, 175-184, 2000.

Sarminingsih, A., Soekarno, I., Hadihardaja, I. K., and Syahril B.K, M.: Flood vulnerability assessment of Upper Citarum

935 River Basin, West Java, Indonesia, International Journal of Applied Engineering Research, 9, 22921-22940, 2014.

Schlehe, J.: Anthropology of religion: Disasters and the representations of tradition and modernity, Religion, 40, 112-120, 10.1016/j.religion.2009.12.004, 2010.

[revised manuscript text omitted]

Van Bemmelen, R. W.: The Geology of Indonesia, General Geology of Indonesia and Adjacent Archipelagoes, Government Printing, The Hague, 1949a.

van Bemmelen, R. W.: Report on the volcanic activity and volcanological research in indonesia during the period 1936–1948, Bulletin Volcanologique, 9, 3-29, 10.1007/bf02596089, 1949b.

van Bemmelen, R. W.: Relations entre le volcanisme et la tectogénèse en Indonésie: Résumé d'un discours pour l'Association Internationale de Volcanologie à Bruxelles, août 1951, Bulletin Volcanologique, 13, 57-62, 10.1007/bf02596791, 1953.

van Bemmelen, R. W.: Volcanology and geology of ignimbrites in Indonesia, North Italy, and the U.S.A, Bulletin Volcanologique, 25, 151-173, 10.1007/bf02596548, 1963.

Van Bemmelen, R. W., and Bourter, E. A. d.: The Geology of Indonesia, General Geology of Indonesia, Government Printing, The Hague, 1970.

van Hinsberg, V., Berlo, K., Sumarti, S., van Bergen, M., and Williams-Jones, A.: Extreme alteration by hyperacidic brines at Kawah Ijen volcano, East Java, Indonesia: II. Metasomatic imprint and element fluxes, Journal of Volcanology and Geothermal Research, 196, 169-184, 10.1016/j.jvolgeores.2010.07.004, 2010.

van Voorst, R.: Formal and informal flood governance in Jakarta, Indonesia, Habitat International, 52, 5-10, 10.1016/j.habitatint.2015.08.023, 2016.

Verstappen, H. T.: Geomorphological surveys and natural hazard zoning, with special reference to volcanic hazards in central Java, Zeitschrift fur Geomorphologie, Supplementband, 68, 81-101, 1988.

Verstappen, H. T.: Volcanic geomorphology and natural disaster reduction - the volcanoes of Indonesia, some examples, Bulletin - Association de Geographes Francais, 1993, 367-376, 1993.

Verstappen, H. T.: The volcanoes of Indonesia and natural disaster reduction (with some examples), Indonesian Journal of Geography, 26, 27-35, 1994.

Vignato, S.: Devices of oblivion: How Islamic schools rescue 'orphaned' children from traumatic experiences in Aceh (Indonesia), South East Asia Research, 20, 239-261, 10.5367/sear.2012.0107, 2012.

Voight, B., Constantine, E. K., Siswowidjoyo, S., and Torley, R.: Historical eruptions of Merapi Volcano, Central Java, Indonesia, 1768-1998, Journal of Volcanology and Geothermal Research, 100, 69-138, 2000.

Wagner, C. S., and Leydesdorff, L.: Network structure, self-organization, and the growth of international collaboration in science, Research Policy, 34, 1608-1618, http://dx.doi.org/10.1016/j.respol.2005.08.002, 2005a.

Wagner, C. S., and Leydesdorff, L.: Mapping the network of global science: comparing international co-authorships from 1990 to 2000, International Journal of Technology and Globalisation, 1, 185-208, 10.1504/ijtg.2005.007050, 2005b.

Ward, P. J., Pauw, W. P., van Buuren, M. W., and Marfai, M. A.: Governance of flood risk management in a time of climate change: The cases of Jakarta and Rotterdam, Environ. Polit., 22, 518-536, 10.1080/09644016.2012.683155, 2013.

Warner, K., van der Geest, K., Kreft, S., Huq, S., Harmeling, S., Kusters, K., and De Sherbinin, A.: Evidence from the frontlines of climate change: loss and damage to communities despite coping and adapation, UNU- EHS, Bonn, 2012.

WFP: Food Security and Vulnerability Atlas of Indonesia, Jakarta, 2015.

Whittaker, J., McLennan, B., and Handmer, J.: A review of informal volunteerism in emergencies and disasters: Definition, opportunities and challenges, International Journal of Disaster Risk Reduction, 13, 358-368, http://dx.doi.org/10.1016/j.ijdrr.2015.07.010, 2015.

2015 is Hottest Year on Record: http://public.wmo.int/en/media/press-release/2015-hottest-year-record, 2016.

Woodhouse, C. A., and Overpeck, J. T.: 2000 Years of Drought Variability in the Central United States, Bulletin of the American Meteorological Society, 79, 2693-2714, 1998.

1070 Zen, M. T., and Hadikusumo, D.: Recent changes in the Anak-Krakatau volcano, Bulletin Volcanologique, 27, 259-268, 10.1007/bf02597525, 1964a.

Zen, M. T., and Hadikusumo, D.: Preliminary report on the 1963 eruption of Mt.Agung in Bali (Indonesia), Bulletin Volcanologique, 27, 269-299, 10.1007/bf02597526, 1964b.

Zen, M. T., and Hadikusumo, D.: The future danger of Mt. Kelut (Eastern Java — Indonesia), Bulletin Volcanologique, 28,
1075 275-282, 10.1007/bf02596932, 1965.

Zen, M. T.: The formation of various ash flows in Indonesia, Bulletin Volcanologique, 29, 77-78, 10.1007/bf02597144, 1966.

Zen, M. T.: Growth and state of Anak Krakatau in September 1968, Bulletin Volcanologique, 34, 205-215, 10.1007/bf02597786, 1970.

1080 Zen, M. T.: Structural origin of Lake Singkarak in central Sumatra, Bulletin Volcanologique, 35, 453-461, 10.1007/bf02596966, 1971.

**List of Tables**

**List of Figures**

---

## Referee Comment (RC2) · Anonymous Referee #2 · 5 Apr 2017

This article is very rich in term of literature and it provides a useful insight for researcher to study DRR and climate change in Indonesia.

---

## Editor Comment (EC1) · B. D. Malamud (Editor) · 11 Apr 2017

Dear Riyanti Djalante:

Thank you for your response to both reviewers. The second review was exceptionally short (it was just one line), and thus does not really constitute a meangiful review. I am therefore seeking a third review. Please be patient while the third reviewer (who has already agreed) comes back with their comments.

Regards, Bruce [NHESS Executive Editor]
* * *
[Figure]

2016.

---

## Author Comment (AC2) · 11 Apr 2017

Dear Reviewer 2. Many thanks for your reply and it is great that you find the article has the potential to enrich the discussions on DRR and CCA in Indonesia.

---

## Short Comment (SC1) · 12 Apr 2017

Currently, this paper appears mainly discuss the history of hazard which happens in Indonesia. Overall, this manuscript is appropriate and can be considered for journal publication. However, I would recommend to several things to be noticed.

1. In the text line 272, we found the citation Subijakto (1992), which is not found in the references. Please check it once again.

2. In line 288, you write the reference for Marfai (2014) and (2015), it is better to choose the print version one, instead of writing it both.

3. It would be more appealing if you can discuss more, about ecosystem bases Disaster Risk Reduction (DRR) and mention about the trend focus of disaster research In Indonesia.

4. The paper should add more information about the importance of collaboration between local universities (Indonesia) and universities partner (outside Indonesia) to promote International publications in Indonesia, particularly to enlarge the topics related to the disaster, hazard, and risk reduction.

5. I would recommend adding references from another peer-reviewed literature except for Scopus, i.e. google scholar, or local database journals from Kemenristek DIKTI Indonesia, such as Shinta and Portal Garuda.

6. The literature could also be taken from the books, proceedings, government's reports. Unfortunately, the government's reports are written in Bahasa Indonesia and not available for the public.

7. To enrich the discussion in this paper, I would suggest searching other keywords associated with disaster and hazard, i.e. erosion, sedimentation, etc.

Based on the things above, I feel that this paper is excellent and could be considered for publication with Minor Revision.

---

## Author Comment (AC3) · 12 Apr 2017

Dear Prof. Marfai,

I would like to thank for your valuable comments which greatly enhance the quality of the recommendations, in particular. Kindly please find my responses below.

Please note that I have made changes to the paper as the response to the first reviewer which is listed in this site http://www.nat-hazards-earth-syst-sci-discuss.net/nhess-2016-342/nhess-2016-342-AC1-supplement.pdf. Hence, my numbering reference to your comments will be based on this revised version.

1. In the text line 272, we found the citation Sudibyakto (1992), which is not found in

the references. Please check it once again.

Response: In the revised version Line 979-981, all references to Sudibyakto 1992, 1996, 1997 are listed.

2. In line 288, you write the reference for Marfai (2014) and (2015), it is better to choose the print version one, instead of writing it both.

Response: Thank you, Marfai(2014) was deleted.

3. It would be more appealing if you can discuss more, about ecosystem bases Disaster Risk Reduction (DRR) and mention about the trend focus of disaster research In Indonesia.

Response: I put this as part of my recommendations: Sentences from Lines 453-456 are revised from:

There is still greater need for research on climate change topics related to linkages between poverty and disaster vulnerability (Suryahadi and Sumarto, 2003), security (CSIS, 2016), loss and damages (Warner et al., 2012), impacts on key sectors such as fisheries (USAID Indonesia, 2015), coastal communities (Marfai, 2014; Marfai et al., 2008), food security (Measey, 2012; WFP, 2015), health (Ady Wirawan, 2010; Haryanto, 2009), migrations (Raleigh et al., 2008; Reuveny, 2007), and community-based DRR (Heijmans, 2012),

to

There is still greater need for research on climate change topics related to linkages between poverty and disaster vulnerability (Suryahadi and Sumarto, 2003), security (CSIS, 2016), loss and damages (Warner et al., 2012), impacts on key sectors such as fisheries (USAID Indonesia, 2015), coastal communities (Marfai, 2014; Marfai et al., 2008), food security (Measey, 2012; WFP, 2015), health (Ady Wirawan, 2010; Haryanto, 2009), migrations (Raleigh et al., 2008; Reuveny, 2007), community-based DRR (Heijmans, 2012), and the role of ecosystem for DRR and CCA (Renaud et al

2016).

4. The paper should add more information about the importance of collaboration between local universities (Indonesia) and universities partner (outside Indonesia) to promote International publications in Indonesia, particularly to enlarge the topics related to the disaster, hazard, and risk reduction.

Response:

Line 469 New sentences are added. There needs to be more collaboration between local universities (Indonesia) and universities partner (outside Indonesia) to promote International publications in Indonesia, particularly to enlarge the topics related to the disaster, hazard, and risk reduction.

5. I would recommend adding references from another peer-reviewed literature except for Scopus, i.e. google scholar, or local database journals from Kemenristek DIKTI Indonesia, such as Shinta and Portal Garuda.

Response: It is not possible to use other databases at this stage. Goggle scholar was indeed used to determine major authors contribution, as listed in Table 3 (Line 345).

However, I add new sentences in Line 463 to acknowledge these databases.

Line 461. The author acknowledges that there are also Indonesia local database from Kemenristek DIKTI Indonesia, such as Shinta and Portal Garuda, and materials from these databases should be made available and accessible internationally.

6. The literature could also be taken from the books, proceedings, government's reports. Unfortunately, the government's reports are written in Bahasa Indonesia and not available for the public.

Response: It is clearly stated in the method section that the source is only from SCO-PUS, which also include books, proceedings and sometimes reports. Those that are not included hence falls outside the scope of this paper.

7. To enrich the discussion in this paper, I would suggest searching other keywords associated with disaster and hazard, i.e. erosion, sedimentation, etc.

Response: Since the keyword 'Hazard' was used, any associated impacts from those hazards would have been included, include erosion and sedimentation, etc. Step 3 of the data inclusion already include these impacts.

Please also note the supplement to this comment:
http://www.nat-hazards-earth-syst-sci-discuss.net/nhess-2016-342/nhess-2016-342-AC3-supplement.pdf

**Supplement:**

**Research Trends in Natural Hazards, Disasters, Risk Reduction and Climate Change in Indonesia: A Systematic Literature Review**

Riyanti Djalante[1, 2]

[1]United Nations University – Institute of Environment and human Security, Bonn, 53117, Germany

5  [2]University of Halu Oleo, Kendari, 93111, Sulawesi Tenggara, Indonesia

*Correspondence to*: Djalante@ehs.unu.edu

**Abstract.** Indonesia is one of the most vulnerable countries from disasters and climate change. While there has been a proliferation of academic publications written on issues related to natural hazards, risks, and disasters on Indonesia, there has not yet a systematic literature review (SLR) to determine the progress, key topics and directions for further research. SLR is

10  important so researchers can build upon existing works, avoid bias, determine major research and need for further research. It is also important to determine who, how, in which way the research has been conducted in order to strengthen research capacity in the future. The author conducted a SLR of publications indexed within the Scopus database from 1900 to 2016 on topics related to natural hazards, risks, risk reduction and climate change impacts on Indonesia. The findings are outlined in two parts. The first part focuses on the research topics and finds that publications can be categorized into three major

15  topics: (1) natural hazard, risk and disaster assessments (HRD), (2) disaster risk reduction (DRR), and (3) climate change risks, vulnerability, impacts and adaptation (CC). More than half the publications fall into HRD and focus on volcanic eruptions, tsunami and earthquakes. Publications on DRR focus on governance, early warning systems and recovery and reconstruction. Those regarding CC mainly concern carbon emission, forestry, governance, and impacts. The second part focuses on roles of Indonesian researchers and organizations in these publications. Findings show limited progress in

20  research, publication and collaboration. International/ non-Indonesian authors dominate the literature and only half of the publications are co-authored by Indonesians. Moreover, of the international collaborations that took place, this was limited to only a few Indonesian organizations. Reasons for this could be limited experience in academic collaboration, power play amongst researchers, lack of research capacity, weak English academic writings skills as well as a lack of incentives for international collaboration and publication within the Indonesian higher education system.

**Keyword: Systematic literature review; natural hazard; disaster; climate change; Indonesia**

**1 Introduction**

Disaster events and their associated social and economical impacts are on the rise (EMDAT, 2016). The last decade has witnessed the highest number and impacts from disasters and 2015 has been declared as the hottest year ever (WMO, 2016). The Asia Pacific region has experienced the highest number of disasters on record (EMDAT, 2016), within which Indonesia is one of the most at risk countries to disasters and climate change impacts (EMDAT, 2016). Between the period of 1900 to 2016, there have been a total of 434 disasters in Indonesia caused by natural hazards, with 237,728 deaths, 29.1 million people affected and total damage almost 30 Billion USD (EMDAT, 2016). Geophysical hazards caused more than 90% deaths while the hydrometerological occur more frequently, affected more people, and caused three times damages (EMDAT, 2016). This paper aims to systematically review literature related to natural hazards, risk and disaster risk reduction, as well as climate change vulnerability, impact, and assessments in Indonesia. A systematic literature review (SLR) is defined as a method for systematically reviewing evidence or literature with explicit and transparent methods (Gill and Malamud, 2014). Even though there is vast material on these topics on Indonesia, there has not yet been a literature review that examines them in a comprehensive and systematic way. By reviewing published works in this fashion, researchers can build upon others' works, avoid bias (Khan et al., 1996) and reinventing the wheel so that topics that have been heavily researched can be determined, and those that need further research can be outlined (Moher et al., 2009b). It is also important to gauge who, how and in which way the research has been conducted, and determining this will enable consideration for strengthening research capacity in the future (Mallett et al., 2012).

There are two research aims adopted. The **first** is to determine progress of research in natural hazards, risks, disasters and climate change in Indonesia within the timeframe from 1900 to 2016. The **second** is to examine roles of Indonesian authors in contributing to research, international publications and collaborations. The importance of conducting literature on these topics is manifold. The Sendai Framework for DRR (SFDRR) has just been adopted and with it an extension of the scope of hazards and risk reduction strategies (UN/ISDR, 2015). There is a move toward an integrated approach to DRR which calls for strategies and actions to reduce risks and associated impacts, as well as an inclusive role of multiple actors in DRR. This review will enable the identification of strategies that have been undertaken for DRR and hence suggest strategies for future DRR and implementing the SFDRR. Also, there is an increasing focus on the impacts of climate change in the changing profile of hazards and disasters, and hence this calls for integrated DRR and climate change adaptation (CCA) to manage climate risks. This review will try to capture whether consideration of climate change risks have been considered as part of research progress in Indonesia. This study attempts to determine whether progress towards more specific studies on the national and local level is observable. Moreover, determining the progress of Indonesian scholars is important and relevant for several reasons. These 
[revised manuscript text omitted]

**2.2 Data Analysis**

The author used  Scopus features to analyze search results such as the article metric module, citation overview, and author profile pages (SCOPUS, 2016b). This final list was analyzed in terms of authorship, references, citations, keywords, places of focus, types and time of publications, impact factors and topics and sub-topics of research. The progress of Indonesian scholars is evaluated through counting total number of authors, research outputs and citations overall, and also comparing between papers first authored by Indonesians. The author cross-checked the number of citations from Scopus on the Internet through Google, and selected the higher citation counts. This was done because it is generally the case that data from a Google search for a publication and author leads to a higher and more up to date citation count. The author also consulted total citations and publications of researchers in Google Scholar, Research Gate or from other websites to make sure that the full list of publications was captured. There were also cases where the author had to specifically go back to Scopus and find particular author's works to make sure that all were captured.

**3 Findings and Analysis**

This section is structured into two main parts, first with research topics, and second with progress of Indonesian researchers and organizations.

**3.1 Timelines and Research Topics**

This part presents the more detailed findings of each of the research topics. The author categorizes the final list into three groups (Table 2), natural hazard, risk, disaster assessments (HRD), disaster risk management and reduction (DRR), and climate change vulnerability, impacts and adaptation (CC), in order to show and outline how changes in directions on research have taken place over the years and to reduce unbalance towards findings on hazard and risks assessments toward earthquake and volcanic eruption research. There are 56% of HRD, and the rest is shared almost equally by the DRR and CC literature (modified from SCOPUS, 2016a).

**Table 2 Classifications of Findings Based on Topics of Research**

| Major topics groups | Definitions (IPCC, 2012; UNISDR, 2009) | Number of publications |
|---|---|---|
| (1) Natural hazard, risks, disasters assessments (HRD) | Hazards: A dangerous phenomenon, substance, human activity or condition that may cause loss of life, injury or other health impacts, property damage, loss of livelihoods and services, social and economic disruption, or environmental damage. Risks: The combination of the probability of an event and its negative consequences. Disaster: A serious disruption of the functioning of a community or a society involving widespread human, material, economic or environmental losses and impacts, which exceeds the ability of the affected community or society to cope using its own resources. | 517 |
| (2) disaster risk management or reduction (DRR) | The systematic process of using administrative directives, organizations, and operational skills and capacities to implement strategies, policies and improved coping capacities in order to lessen the adverse impacts of hazards and the possibility of disaster (UNISDR). The concept and practice of reducing disaster risks through systematic efforts to analyze and manage the causal factors of disasters, including through reduced exposure to hazards, lessened vulnerability of people and property, wise management of land and the environment, and improved preparedness for adverse events. | 210 |
| (3) climate change vulnerability, impacts and adaptation (CC) | A change of climate which is attributed directly or indirectly to human activity that alters the composition of the global atmosphere and which is in addition to natural climate variability observed over comparable time periods (UNFCCC). The adjustment in natural or human systems in response to actual or expected climatic stimuli or their effects, which moderates harm or exploits beneficial opportunities (UNISDR). | 194 |
| **Total** | | 921 |

The paper identifies key periods and timelines by which publications were published. In general, there are more research on the topic of HRD, followed by those in DRR, and then CC. The publications on the HRD are also some of the earliest publications indexed in Scopus. Although the search timeline was set between 1900 and 2016, the years in which publications were found ranges from 1934 to 2016 (Figure 1).

[Figure]

Figure 1 Number of Publications over the Year (modified from SCOPUS, 2016a)

The first period is from the 1934-1990s. There were no significant changes in the numbers of publications produced. Research in this period was heavily focused on the topics of geophysical hazards and risks related to earthquakes and volcanic eruptions  (SCOPUS, 2016a). Within this period, 22 out of 58 events recorded by EMDAT were earthquakes and volcanic activities (EMDAT, 2016). The Bali earthquakes occurred in 1976 and 1979, which in total caused 1764 deaths, affected 563,150 people, and caused 215,150 USD in damages (EMDAT, 2016). The year 1979 was also the year in which the earthquake occurred the most (6 times), in Bali, Lombok, and Biak (near Papua) (USGS, 2016). The second period from the 1990s to 2000s shows a notable increase in the literature, up to an average there 10 publications per year. This gradual increase mainly corresponds to a rise in literature related to the assessments of hazards, risks and disasters, and is followed by a sharp increase in literature to its highest point in 2000 (SCOPUS, 2016a). The third period from 2000s-2010s was the most dynamic period for publications. While there was a sharp decline since it first peak in 2000, a surge of publications begun in 2004 in response to the Indian Ocean tsunami which devastated Indonesia especially. This increase has continued ever since. This is also a period characterized not only publications related to understanding the risks of earthquakes and tsunami, but also those related to DRR and CC. A peak occurs between 2010 and 2016 which shows soaring published materials in all topics. There were 153 publications in 2016 which is the highest ever produced in a single year. During this period, publications related to climate change and its impact on Indonesia has started to be considered and is expected to rise further in the future. Both publications on HRD and CC are expected to rise (SCOPUS, 2016a).

The following sub-sections outline research issues discussed within the three topic groups. Within each, the paper discusses timelines, focus areas of the research, early contributors, and categorization of key topics discussed.

**3.1.1 Natural hazards, risks and disasters assessments (HRD)**

The first sub-section explains findings on the topic of hazards, risks and disasters assessments and identifications. The EMDAT-CRED (2016) categorization of HRD that is used in this study to help more detailed analysis related to major research topics. Natural-disaster groups caused by geophysical, meteorological, hydrological, and climatologically hazards are included since it is determined that these are the most frequent and impactful disasters in the country. Those excluded are disasters caused by biological, extraterrestrial and technological hazards.

There are 535 publications in this category (SCOPUS, 2016a). The findings show that there has been a gradual increase in the number of published materials from 1934 to 2000. It first reached its first peak in 2000 that the research in this topic reached its first significant outputs of 25 publications, and reduced slightly after that. In 2004 the Indian Ocean tsunami occurred, initiated with the 9.8 M earthquake with the epicenter off the island of Sumatra, badly affecting Indonesian. Publications related to the tsunami continued to be published until it reached a peak in 2006. Then in 2009, the publications have increased rapidly ever since, reaching another peak in 2015 of 47 publications in a single year (SCOPUS, 2016a). The islands of Java and Sumatera are the two areas which receive most attention (more than 70%) (SCOPUS, 2016a). The studies in these two islands are mostly related to the study of volcanic eruptions, earthquakes and tsunami. This is not surprising considering that Indonesia has the most numbers of volcanoes and is located along the Pacific ring of fire where earthquakes occur the most (USGS, 2016). The island of Sumatera directly experienced and was impacted by one of the most powerful earthquakes of 8.9 R.S which caused the tsunami in 2004 and hit Aceh, in the north west of Sumatera (Ishii et al., 2005).

Most of the literature around this period focuses on the impacts of volcanic eruptions in Java and Sumatera. The oldest publications related to HRD in Indonesia 
[revised manuscript text omitted]
. Studies on the roles of international and local authorships and collaborations show that although it rapidly increasing, there are still more efforts needed to strengthen and advance those collaborations (Bordons et al., 1996; Wagner and Leydesdorff, 2005b, a; Gazni et al., 2012). It further shows that there is still imbalance in the ratio of male to female scientists, as the global trends also show (Sidhu et al., 2009; Lewison, 2001; Koppel et al., 2002; Sugimoto et al., 2013). The importance of science communication and the increasing demand for researchers to publish their works outside of traditional methods such as journal articles, but also through blogs, websites, policy briefs, and popular media is now encouraged (Gu and Widén-Wulff, 2011; Thelwall et al., 2013; Bik and Goldstein, 2013).

**3.2.1 Authorships**

[revised manuscript text omitted]

**Acknowledgment**

The author would like to acknowledge her Alexander von Humboldt Fellowship for Experienced Researchers which facilitates her research in Germany at the United Nations University Institute for Environment and Human Security. The author benefits enormously from the reviewers' comments and has greatly improved to quality of the paper.

1100 **List of Tables**

**List of Figures**

---

## Short Comment (SC2) · 20 Apr 2017

Dear Riyanti,

I appreciate your response to my recommendation. This paper is ready to be published after the revisions you've made. Congratulations!

---

## Referee Comment (RC3) · Anonymous Referee #3 · 2 May 2017

**Review on "Research Trends on Natural Hazards, Disasters, Risk Reduction and Climate Change in Indonesia: A Systematic Literature Review" by Riyanti Djalante**

Nat. Hazards Earth Syst. Sci. Discuss.
doi: 10.5194/nhess-2016-342-RC1, 2016

**General Comments:**
The paper is a valuable contribution, especially to the researchers from Indonesia. It provides insights and indicates new trends for natural hazards, disaster risk reduction, and climate change research in one of the most high disaster risks profile in Asian region, which motivate and trigger Indonesian researchers to write and publish more of their findings within above 3 scopes of research in the international arena. Please allow me to convey specific comments towards the betterment of the paper below.

**Specific Comments:**

| Sequence and Content | Deliverance, Usefulness, and Lessons Learned
*Input regarding the view on the content of the lessons/case studies from the point of view of advantages or usefulness to the readers.* | Methodology
*Input regarding on the suitability of the writing with reference to the standardized writing rules, including the presentation of quotations, case studies, references, etc.* |
|---|---|---|
| **Storyline (logical order)**

▪ The paper has been presented in the manner, such as: starting from illustrating the different types of natural hazards and risks impacting Indonesia and the comparison between geophysical and hydro-meteor-climato-logical disaster (Line 33-35) (including the graph to underline and distinguished that comparison) and subsequently describing the aims, research questions, advancement of methods, analysis, and conclusion.

▪ However, personally, to make this paper a perfect one; the author could link the missing piece of thesis statement or state stronger the relevance between pinpointing the | ▪ No doubt, the paper is well formulated, rich with new insights with vast literature, as well as the paper is extremely important for the readers, especially for the Indonesian researchers to take up new information, suggestions, and recommendations from the author. The Indonesian researcher could reflect, and set new courses of researches in terms of HRD, DRR, and CC, for example in areas where there are still huge gaps, according to author, in terms of less number of Indonesian contributions as first author, limited number of Indonesian organizations participate in international collaborations, insufficient power play amongst researchers, research capacity, English academic writing, and incentives. | In my personal opinion, the writing of the paper is of high quality. The author has used complicated method and required high focus and vast amount of time. The presentation of the quotations and sources of literature have been mentioned throughout the text and in the list of references. Due to vast amount of used references, it is worth to double-check the list in the end of paper, to avoid discrepancies. |

comparison of geophysical and hydro-meteor-climato-logical disaster and the aims of the paper. The author could formulate stronger statement(s) of why elaborating the different disasters' impacts in the first place (as the intro) and later the aim(s) of the review. Please kindly state stronger motivation(s) of why reviewing natural hazards-DRR-and CC literatures in Indonesia with supporting references.

**Content's proportion**
- The proportion of the content is illustrated very well. Minor suggestion would be in every section of HRD, DRR, and CC, especially in the section of Finding and Analysis; it would be better to elaborate more on the timeline, discussions, and focus area part.
- It would be even better to have a summary/overview table of key findings and analysis with x-axis are the HRD, DRR, and CC and y-axis are the timeline, discussions, and focus area.

**Consistency of used terms and accuracy**
- Please kindly check the used terms of hydro-meteor-climato-logical disaster (Line 33), whether it is the correct writing? And whether the above term is in line and consistent with the later used term throughout the text? For example in Line 176-177, Line 188, and Line 432.
- The used term risk reduction maybe is a minor typo as risk deduction in Line 505-506.
- It would be better to spell out/introduce the abbreviations and acronyms used in the paper for the first time (within parenthesis) and

- However, it would even better useful, especially for Indonesian researchers if the author could suggest and explore concrete key ideas and how to transform those ideas into practical actions (not only referring the setbacks) i.e. to conduct better/improve research, negotiate for authorship amongst international researchers, overcome challenges in international collaborations, factors contributing in improving research capacities including academic writing in English, and innovation of some sort of incentives for international collaborations and publications. These might very useful to the readers and add precious value to the paper.

| | | |
|---|---|---|
| later on only mention the short term. | | |
| **Content's structure**
▪ The content of the structure in this paper is well organized. Minor suggestions would be on the content's proportion and consistency of the used terms (please kindly see the comments for the proportion and structure boxes). | | |
| **Inter-linkages between parts**
▪ The link between different parts is strongly shown in the writing; although stronger link is needed to rationalize the purpose of the paper and the written different impacts of different type of disasters that were stated in the introduction part. | | |
| **Content vs. Theme**
▪ In my opinion, the content of this paper is generally justified with the overarching theme. Minor adjustments would be only on the aforementioned comments. | | |

---

## Author Comment (AC6) · 4 May 2017

Dear Prof. Malamud.

Please be advised that I have responded to the comments from Reviewer No. 3. The review was very detailed and I benefited a lot from the comments especially on strategies to improve the quality of future research and scientific writings of the Indonesian researchers.

Best Regards,

Riyanti Djalante

---

## Author Comment (AC7) · 8 May 2017

Dear Prof. Malamud.

Please be advised that I have responded to all reviewers'comments for this article. One common theme that was raised the reviewers'interest and hence reflected in their comments, was the roles of Indonesian researchers within these publications included in the literature review. Hence I kindly request to change the tittle of the paper to be:

'Research Trends and Authorship on Natural Hazards, Disasters, Risk Reduction and Climate Change in Indonesia: A Systematic Literature Review'

This new tittle add two words of 'and Authorship' to the original tittle.

[Figure]

Best Regards,

Riyanti Djalante

---

## Author Response (AR1)

Dear Reviewer, The author would like to say thank you for the valuable comments given on this paper.

Kindly please find detailed responses to each issue raised in the review:

1. Comment: Abstract: The timeline considered isn't it from 1977 as expressed later in the manuscript (line 147; pg 6)?

Response: The sentence is revised Line 10: To address this, the author conducted a systematic literature review of related publications indexed within the SCOPUS database from 1900 to 2016.

[Figure]

2. Comment: The author reviews three main topics (1) HDR (2) DRR and (3) CC, but the introduction just focuses on the presentation of disaster events in Indonesia and on the comparison between geophysical and hydro-meteor-climato-logical disasters (is this last word correct by the way?). This small introduction (lines 24-35) does not give a clear picture of why the author undertook this review.

Response: Line 32: The aim is rewritten This paper aims to systematically review literature related to natural hazards, risk and disaster risk reduction, along with the strategies to mitigate and manage the events and impacts, as well as climate change vulnerability, impact, and assessments in Indonesia.

3. Comment: I suggest to report the information of figure 2 directly in the text and therefore to remove figure 2.

Response: Figure 2 is removed

4. Comment: Furthermore, after recognizing the gaps of knowledge, the author can present the objectives of the paper. Accordingly, I suggest to finish the introduction section with "this papers aims to systematically review.." that is presented from line 38 onwards in pg.2. Response: Line 30-55 show the elaborations on the aim of the paper.

5. Comment: Research method: The methodology has been undertaken correctly since few researchers explore the systematic literature review, as it is complex and time consuming. Anyway, the sub-chapter 2.1, 2.2, 2.2.1, 2.2.2 and 2.2.3 should be merged together into one main body under "2. Research methods". The chapter 2.3 "Analysis and presentation of results" seems a repetition of the small subchapter presented before for which I suggest merging this in the previous chapter.

Response: The author reorganize this section completely, and to have only sub-headings 2.1 on Data Collection, and and 2.2 on Data Analysis.

6. Comment: Findings and Analysis: The categorization of disaster groups in table 4 is taken from EM-DAT 2016 and any added value is given. I suggest removing the table

and integrating the citation in the text. When the author presents each topic (3.1.1-onwards), there is no need to explore them with so many small subchapters (timeline, discussions and focus areas). I suggest to merge these paragraphs trying to give an overall sequence and shape. The same for the topics 3.3.2 and 3.3.3. I think that there are some errors in the numbering of the chapters. Please revise it carefully. The discussion part presented from line 285 to 297 pg. 10-11 is not discussed at all. Please provide some ideas and key points on it. In its form it is a mere list.

Response: The author also simplified this section, and the small subchapters are merged.

6. Comment: The "Progress of Indonesian researchers and organization" chapter is valuable and I personally think that is the core of the paper, never explored in literature before. However, I think that this chapter could be presented without so many sub-chapters and the author needs to rearrange it merging the information into one main section. Please, try to present the results of this chapter without coping and pasting the paragraphs in chronologically way they are presented now. In addition, the text in lines 328-334 seems an advertisement of first authors as expressed in table 5. I would delete this information or rearranged the way it is presented.

Response: Section 3.2 is also simplified.

6. Table 5. Please report in the caption all the acronyms used (e.g PVMBG, ITB etc). I cannot understand the symbol put after SC, GS, RG. In addition, I would delete the column field "other profile" since does not give any added value. Does the author Surono have a name?

Response: All acronims have been spelt out when they first used. Column on other profile is deleted Surono is a one name author as stated in Scopus

Please also note the supplement to this comment:
http://www.nat-hazards-earth-syst-sci-discuss.net/nhess-2016-342/nhess-2016-342-

[Figure]

AC1-supplement.pdf

[Figure]

Nat. Hazards Earth Syst. Sci. Discuss.,
doi:10.5194/nhess-2016-342-AC2, 2017

[Figure]

Dear Reviewer 2. Many thanks for your reply and it is great that you find the article has the potential to enrich the discussions on DRR and CCA in Indonesia.

[Figure]

Nat. Hazards Earth Syst. Sci. Discuss.,
doi:10.5194/nhess-2016-342-AC4, 2017

[Figure]

I had just uploaded my latest changes based on 2 reviewers'reviews and 1 short comments from Prof. Marfai. He is one of the most prominent researchers on this topic (identified in this paper). I had also presented this paper at the UNU-TWIN SEA workshop by which key authors on the topics on DRR, CCA in Indonesia were present. The researchers represent those from ITB, LIPI, UGM and also those involved in UNU-EHS collaboration with LIP through TWIN-SEA network. They generally accept the findings and stated that the paper had been able to capture the progress for DRR research in Indonesia.

[Figure]

The link to TWIN-SEA is https://ehs.unu.edu/research/twin-sea-expert-network-and-twinning-institute-on-climate-and-societal-change-for-southeast-asia.html#outline.

Best regards, Riyanti
* * *
[Figure]

Nat. Hazards Earth Syst. Sci. Discuss.,
doi:10.5194/nhess-2016-342-AC3, 2017

[Figure]
I would like to thank for your valuable comments which greatly enhance the quality of the recommendations, in particular. Kindly please find my responses below.

Please note that I have made changes to the paper as the response to the first reviewer which is listed in this site http://www.nat-hazards-earth-syst-sci-discuss.net/nhess-2016-342/nhess-2016-342-AC1-supplement.pdf. Hence, my numbering reference to your comments will be based on this revised version.

1. In the text line 272, we found the citation Sudibyakto (1992), which is not found in

the references. Please check it once again.

Response: In the revised version Line 979-981, all references to Sudibyakto 1992, 1996, 1997 are listed.

2. In line 288, you write the reference for Marfai (2014) and (2015), it is better to choose the print version one, instead of writing it both.

Response: Thank you, Marfai(2014) was deleted.

3. It would be more appealing if you can discuss more, about ecosystem bases Disaster Risk Reduction (DRR) and mention about the trend focus of disaster research In Indonesia.

Response: I put this as part of my recommendations: Sentences from Lines 453-456 are revised from:

There is still greater need for research on climate change topics related to linkages between poverty and disaster vulnerability (Suryahadi and Sumarto, 2003), security (CSIS, 2016), loss and damages (Warner et al., 2012), impacts on key sectors such as fisheries (USAID Indonesia, 2015), coastal communities (Marfai, 2014; Marfai et al., 2008), food security (Measey, 2012; WFP, 2015), health (Ady Wirawan, 2010; Haryanto, 2009), migrations (Raleigh et al., 2008; Reuveny, 2007), and community-based DRR (Heijmans, 2012),

to

There is still greater need for research on climate change topics related to linkages between poverty and disaster vulnerability (Suryahadi and Sumarto, 2003), security (CSIS, 2016), loss and damages (Warner et al., 2012), impacts on key sectors such as fisheries (USAID Indonesia, 2015), coastal communities (Marfai, 2014; Marfai et al., 2008), food security (Measey, 2012; WFP, 2015), health (Ady Wirawan, 2010; Haryanto, 2009), migrations (Raleigh et al., 2008; Reuveny, 2007), community-based DRR (Heijmans, 2012), and the role of ecosystem for DRR and CCA (Renaud et al

2016).

4. The paper should add more information about the importance of collaboration between local universities (Indonesia) and universities partner (outside Indonesia) to promote International publications in Indonesia, particularly to enlarge the topics related to the disaster, hazard, and risk reduction.

Response:

Line 469 New sentences are added. There needs to be more collaboration between local universities (Indonesia) and universities partner (outside Indonesia) to promote International publications in Indonesia, particularly to enlarge the topics related to the disaster, hazard, and risk reduction.

5. I would recommend adding references from another peer-reviewed literature except for Scopus, i.e. google scholar, or local database journals from Kemenristek DIKTI Indonesia, such as Shinta and Portal Garuda.

Response: It is not possible to use other databases at this stage. Goggle scholar was indeed used to determine major authors contribution, as listed in Table 3 (Line 345).

However, I add new sentences in Line 463 to acknowledge these databases.

Line 461. The author acknowledges that there are also Indonesia local database from Kemenristek DIKTI Indonesia, such as Shinta and Portal Garuda, and materials from these databases should be made available and accessible internationally.

6. The literature could also be taken from the books, proceedings, government's reports. Unfortunately, the government's reports are written in Bahasa Indonesia and not available for the public.

Response: It is clearly stated in the method section that the source is only from SCOPUS, which also include books, proceedings and sometimes reports. Those that are not included hence falls outside the scope of this paper.

[Figure]

7. To enrich the discussion in this paper, I would suggest searching other keywords associated with disaster and hazard, i.e. erosion, sedimentation, etc.

Response: Since the keyword 'Hazard'was used, any associated impacts from those hazards would have been included, include erosion and sedimentation, etc. Step 3 of the data inclusion already include these impacts.

Please also note the supplement to this comment:
http://www.nat-hazards-earth-syst-sci-discuss.net/nhess-2016-342/nhess-2016-342-AC3-supplement.pdf

[Figure]

Nat. Hazards Earth Syst. Sci. Discuss.,
doi:10.5194/nhess-2016-342-AC5, 2017

[Figure]

I would like to thank for your detailed assessment of the paper. I am delighted that you find the paper is valuable and contributes to the betterment and increased roles of Indonesian researchers in the future.

The review was extremely detail discussing on the terminologies, formats, references and abbreviations, and this has greatly improved the readability of the paper.

I greatly value your suggestions on focusing and outlining strategies on improving the authorship and quality of publications for Indonesian authors / researchers.

[Figure]

Within the Supplement, I list 2 documents:

1. My responses which outline point-by-point response to your review.

2. The revised version of the paper, by which the references on line numbers as stated in document 1.

Best Regards,

Author.

Please also note the supplement to this comment:
http://www.nat-hazards-earth-syst-sci-discuss.net/nhess-2016-342/nhess-2016-342-AC5-supplement.zip

––––––––––––––––––––––––––––

[Figure]

**Review Article: Research trends in natural hazards, disasters, risk reduction and climate change in Indonesia - a systematic literature review**

5   Riyanti Djalante [1,2]

[1]United Nations University – Institute of Environment and human Security, Bonn, 53117, Germany
[2] University of Halu Oleo, Kendari, 93111, Sulawesi Tenggara, Indonesia

*Correspondence to*: Riyanti Djalante (djalante@ehs.unu.edu)

10   **Abstract**[DR1]. ~~Indonesia is one of the most vulnerable countries from disasters and climate change. There have also been extensive literatures on the examinations of hazards, risks, disasters and disaster risk reduction strategies on Indonesia. To determine progress on those researches, the author conducts a systematic literature review on these topics. A multi-staged review was conducted to study publications that are indexed within SCOPUS with the timeline from 1900 to 2016.~~

15   ~~The findings are outlined in two parts. The first part focuses on the research topics. The publications can be categorized into three major topics of (1) hazard, risks and disaster assessments (HRD), (2) disaster risk reduction (DRR), and (3) climate change risks, vulnerability, impacts and adaptation (CC). The oldest publication was in 1978.Publications on HRD are comprised of more than half of the total publications, focusing on volcanic eruption, tsunami and earthquake. Research on DRR focuses on governance, recovery and reconstruction, early warning systems. Those on CC are mainly on emission~~
20

[revised manuscript text omitted]

---

## Author Response (AR2)

**Review Article: a systematic literature review of research trends and authorships on natural hazards, disasters, risk reduction and climate change in Indonesia**

Riyanti Djalante

**RESPONSES TO THE EDITOR:**

Dear Prof. Heidi Kreibich,

I would like to thank you for the opportunity to resubmit my article on the above title. I have endeavoured to revise the paper adding more descriptions to strengthen the "story" and "narratives" throughout and from the introduction to the conclusion. I have added substantial text in the second paragraph on the introduction to explain the context and rationale for the study. I have also revised the structure slightly so that the outcome of the paper is better explained in the recommendations and concluding part.

In particular:

(i)      Section 4 on recommendations is rewritten to strengthen the 2 main recommendations made. The first one focusses on outlining future research topics including those on risk assessment and climate adaptation in Indonesia. The second recommendation is also rewritten in outlining the need for incentives for increasing the capacity of Indonesian researchers.

(ii)     The background and framing are strengthened through adding a paragraph in section 1, Introduction. More description and discussions on the findings are added.

**RESPONSES TO THE REVIEWERS COMMENTS:**

Dear Reviewer 1,

Thank four for your time in reviewing this article. In section 4 on Recommendations, I have added some discussions on types of policies that are relevant for various planning and implementing agencies on addressing climate change.

5   Page 13 line 15-20, line 16-32, page 14 line 1 -16, page 15 line 16-21. There are 3 paragraphs calling for government intervention on assessments of different climatic risks, understanding the impacts on different development sectors, and addressing the most vulnerable regions and communities.

**RESPONSES TO THE REVIEWERS COMMENTS:**

10   Dear Reviewer 2,

Thank you so much for your comprehensive and detailed reviews. Please find my detailed responses:

- Abstract: Line 11-13 is revised. The revised sentence is "While there has been a proliferation of academic publications on natural hazards, risks, and disasters on Indonesia, there has not yet a systematic literature review (SLR) to determine the progress, key topics, authorships and directions for further research".

15   - Introduction: I have added a whole paragraph (lines 11-21) which outlines a more detailed, structured and robust introduction to the topic. The sentences are "Studies on disasters have expanded enormously globally including in Indonesia and hence there needs to be frequent reviews that examine research trends and topics, issues, challenges and strategies and innovations in dealing with those disasters. The role of science in influencing DRR policy is gaining recognition and studies are needed to examine and identify key lessons learnt and effectiveness of those policies. There is

20   also call for giving more voices and strengthening capacities of local scientists in contributing to the generation of and synthesis of knowledge about their countries. It is often the case that local scientists are being left out in the internationally research collaborations and research publications. The global progress on scholarly publications on disaster science is documented in a report by Elsevier in 2017 on 'A Global Outlook on Disaster Science (Elsevier, 2017). It looks at scholarly outputs and impacts of disaster science according to the SFDRR, and documents progress of several countries in terms of

25   their productivity in producing scholarly studies on disasters. It is found that that are 27,273 outputs which represent only 0.22% of the world's total output, with countries such as China, USA and Japan dominate. Indonesia is part of the 7 countries that specialized in disaster science than global average (Elsevier, 2017). A detailed study that looks at the progress of research and roles of researchers in Indonesia us not yet available."

- Data collection:

30   - The author rearranges the structure of section 2.1 page 3 line 17-26. This is to provide better explanations on the inclusion and exclusions criteria in all three steps.

      o  Page 3 lines 13-22: The sentences were added in the past to respond to the editor comments, Prof Malamud, on giving examples on the use of systematic literature review (SLR) on particular hazards or issues.

      o  Page 3 lines 23-29. The sentences are shortened.

- o Page 3 line 20. The explanation to use the keyword geology is given
- o Page 3 line 20-21. The sentence is revised.
- o Page 3 line 24. The word third is removed.
- Findings and analysis
- o Page 4 line 13. The heading is changed to Findings and Discussions
- o Page 4 line 5. The sentence is removed (page 4 line 29)
- o Page 5 line 25 caption. These numbers are removed. The figure is revised. 153 is the total number in 2016. 921 is the total publication considered.
- o Page 6 line 8, and all others. Citation of Scopus are deleted.
- o **Reviewer 2 raises some very important comments on outlining hot topics, most important arguments, least represented, and explaining why the results as they are. The following sentences are added in section 3.1.1, 3.1.2, 3.1.3**
- o Page 6 line 1-2. Others on Figure 2 is explained
- o Section 3.1.1, Page 6 line 4-18. A paragraph is added for discussions explaining trends, most important discussions, and topics that need further research.
- o Section 3.1.2, Page 7 line 13-20. A paragraph is added for discussions explaining trends, most important discussions, and topics that need further research.
- o Section 3.1.3, Page 8 line 11-19. A paragraph is added for explaining water issues in the figure
- o Section 3.1.3, Page 8 line 26-35. A paragraph is added for discussions explaining trends, most important discussions, and topics that need further research.
- o Page 6 line 27-30, page 7 line 21-30, page 8 line 31-33. The sentences are moved to the section on authorships (3.2.1)
- o Section 3.2.1. page 9 line 8 to page 10 line 7. Discussion on earliest and most important publications on the three topics are moved here.
- Recommendations
- o Page 13 line 15-20 and line 27-page 14 line 16) The recommendations are rewritten to reflect the need to address governments efforts for CCA, and also more details explanation on the recommendations.
- o Page 15 line 16 -21 the concluding remarks are revised.
- Figures and tables
- o Table 1, a column on description is added for better clarification page 30 and 31
- o Table 2, UNFCCC is deleted, instead IPCC is used
- o Table 3. The whole table is rechecked and revised.
- o Figure numbers are revised
- o Figure 1 is revised

- o Figure 2 on other (explanation added)
- o Figure 3, cbdrr is explained (community-based disaster risk reduction)
- o Figure 4, issues on water is explained in the text
- o Figure 5, the figure is revised.
- o Figure 6, the figure is revised.
- o Figure 7, the figure is revised.
- o

[revised manuscript text omitted]

---

## Author Response (AR3)

**Review Article: a systematic literature review of research trends and authorships on natural hazards, disasters, risk reduction and climate change in Indonesia**

Riyanti Djalante [1,2,3]

5   [1] United Nations University – Institute of Environment and human Security, Bonn, 53117, Germany
[2] United Nations University – Institute for the Advanced Study of Sustainability, Tokyo, 150-8925, Japan (Current Affiliation)
[3] University of Halu Oleo, Kendari, 93111, Sulawesi Tenggara, Indonesia

*Correspondence to*: Riyanti Djalante (djalante@unu.edu)

10   **RESPONSES TO THE EDITOR:**

Dear Prof. Heidi Kreibich,
I would like to thank you for accepting the paper. The paper is rechecked and now ready for final publication.

15   Best Regards,

Riyanti